# Impact of mineral dust on the global nitrate aerosol direct and indirect radiative effect

Alexandros Milousis[1], Klaus Klingmüller[2], Alexandra P. Tsimpidi[1], Jasper F. Kok[3], Maria Kanakidou[4,5,6], Athanasios Nenes[5,7], and Vlassis A. Karydis[1]

[1]Institute of Climate and Energy Systems: Troposphere (ICE-3), Forschungszentrum Jülich GmbH, Jülich, Germany
[2]Max Planck Institute for Chemistry, Mainz, Germany
[3]Department of Atmospheric and Oceanic Sciences, University of California Los Angeles, Los Angeles, CA, USA.
[4]Environmental Chemical Processes Laboratory, Department of Chemistry, University of Crete, Heraklion, Greece
[5]Center for the Study of Air Quality and Climate Change, Foundation for Research & Technology Hellas, Patras, Greece
[6]Institute of Environmental Physics, University of Bremen, Bremen, Germany
[7]Laboratory of Atmospheric Processes and Their Impacts, Ecole Polytechnique Fédérale de Lausanne, Switzerland

*Correspondence to*: Vlassis A. Karydis (v.karydis@fz-juelich.de)

## Abstract

Nitrate ($NO_3^-$) aerosol is projected to increase dramatically in the coming decades and may become the dominant inorganic particle species. This is due to the continued strong decrease in $SO_2$ emissions, which is not accompanied by a corresponding decrease in $NO_x$ and especially $NH_3$ emissions. Thus, the radiative effect (RE) of $NO_3^-$ aerosol may become more important than that of $SO_4^{2-}$ aerosol in the future. The physicochemical interactions of mineral dust particles with gas and aerosol tracers play an important role in influencing the overall RE of dust and non-dust aerosols but can be a major source of uncertainty due to their lack of representation in many global climate models. Therefore, this study investigates how and to what extent dust affects the current global $NO_3^-$ aerosol radiative effect through both radiation ($RE_{ari}$) and cloud interactions ($RE_{aci}$) at the top of the atmosphere (TOA). For this purpose, multi-year simulations nudged towards the observed atmospheric circulation were performed with the global atmospheric chemistry and climate model EMAC, while the thermodynamics of the interactions between inorganic aerosols and mineral dust were simulated with the thermodynamic equilibrium model ISORROPIA-lite. The emission flux of the mineral cations $Na^+$, $Ca^{2+}$, $K^+$ and $Mg^{2+}$ is calculated as a fraction of the total aeolian dust emission based on the unique chemical composition of the major deserts worldwide. Our results reveal positive and negative shortwave and longwave radiative effects in different regions of the world via aerosol-radiation interactions and cloud adjustments. Overall, the $NO_3^-$ aerosol direct effect contributes a global cooling of -0.11 $W/m^2$, driven by fine-mode particle cooling at short wavelengths. Regarding the indirect effect, it is noteworthy that $NO_3^-$ aerosol exerts a global mean warming of +0.17 $W/m^2$. While the presence of $NO_3^-$ aerosol enhances the ability of mineral dust particles to act as cloud condensation nuclei (CCN), it simultaneously inhibits the formation of cloud droplets from the smaller anthropogenic particles. This is due to the coagulation of fine anthropogenic CCN particles with the larger nitrate-coated mineral dust particles, which leads to a reduction in total aerosol number concentration. This mechanism results in an overall reduced cloud albedo effect and is thus attributed as warming.

**Keywords:** direct radiative effect, indirect radiative effect, nitrate aerosols, mineral dust

# 1. Introduction

Atmospheric aerosols are among the most complex components of the Earth's climate system. This is due not only to the diversity of their origins, with many natural and anthropogenic emission sources, but also to their extremely varied chemical composition and properties. The many mechanisms by which they interact with each other and with physical entities such as radiation, clouds, land, and oceans add to their complexity and play a critical role in the energy balance of the planet (Arias et al., 2021). The most direct way in which aerosols affect the Earth's energy balance is through their interactions with solar shortwave (SW) and terrestrial longwave (LW) radiation (IPCC, 2013). Overall, the radiative effect due to aerosol-radiation interactions ($RE_{ari}$) is mainly dominated by the scattering of SW radiation back to space (negative radiative effect, generating a cooling of the climate system) and the absorption of LW radiation (positive radiative effect, generating a warming of the climate system) (Gao et al., 2018; Tsigaridis and Kanakidou, 2018). Aerosols belonging to the black and/or brown carbon family, together with mineral dust particles, contribute to absorption (Kanakidou et al., 2005; Zhang et al., 2017; Wong et al., 2019), while the main inorganic aerosol components, such as sulfate and nitrate, as well as a significant amount of organic carbon contribute mainly to scattering (Kirchstetter et al., 2004; (Bond and Bergstrom, 2006; Klingmüller et al., 2019; Zhang, 2020). However, mineral dust can also influence the behavior of the $RE_{ari}$ of anthropogenic pollution. Dust particles alter the anthropogenic radiative effect of aerosol-radiation interactions by reducing the loading of anthropogenic aerosols (either by coagulating with them or by adsorption of their precursor inorganic trace gases), leading to less scattering of solar radiation and thus a warming effect (Kok et al., 2023).

Atmospheric aerosols can also indirectly affect the Earth's energy balance by forming clouds, controlling cloud optical thickness and scattering properties, and altering their precipitation and lifetime (IPCC, 2013). Atmospheric aerosols act as cloud condensation nuclei (CCN), providing a suitable surface for water vapor to condense, leading to the formation of liquid droplets that develop into a corresponding liquid cloud (Lance et al., 2004). Such clouds are referred to as warm clouds and are typically found in the lower troposphere (Khain and Pinsky, 2018). However, there is constant competition between small and large particles for the available amount of water vapor (Barahona et al., 2010; Morales and Nenes, 2014). Under the same humidity conditions, the presence of small particles will lead to the formation of small droplets with high number concentrations, while the presence of larger particles will lead to the formation of large droplets but with lower number concentrations. Depending on the size characteristics of its particle population, a warm cloud will exhibit different optical properties, with a population dominated by smaller particles generally being more reactive in the SW spectrum. The change in cloud reflectivity due to the presence of aerosols is referred to as the first radiative effect due to aerosol-cloud interactions ($RE_{aci}$) and was first described by Twomey (1977). The small size of anthropogenic aerosols results in an overall smaller cloud droplet size, which reduces precipitation efficiency and thus increases cloud lifetime. This contributes to cloud reflectivity and is referred to as the second radiative effect of aerosol cloud-interactions, first described by Albrecht (1989). These two indirect effects are considered equally important for the total indirect radiative effect of aerosols (Lohmann and Feichter, 2005). Atmospheric aerosols exert a net cooling effect that can partially mask the warming effect of greenhouse gases, therefore, the recent decline in

anthropogenic aerosol concentrations may accelerate global warming (Urdiales-Flores et al.,
2023). Overall, the radiative effect due to aerosol-cloud interactions is considered the main source
of existing uncertainty in the effective (total) radiative effect of aerosols in the atmosphere (Myhre
et al., 2014; Seinfeld et al., 2016).
Mineral dust influences the anthropogenic radiative effect through aerosol-cloud interactions in
several ways that can result in either a net warming or net cooling effect. Dust particles can increase
the of cloud droplet number concentrations (CDNC) in remote areas since through chemical aging
by pollutants (Nenes et al., 2014; Karydis et al., 2017), dust particles become more hygroscopic
and require lower supersaturation thresholds for activation (Karydis et al., 2011). This is caused
by the transfer of anthropogenic pollutants towards remote desert regions which enhances the
solubility of dust particles. In such regions, this mostly results in increased cloud albedo and a net
cooling effect. However, dust particles also tend to reduce the availability of smaller anthropogenic
CCN. This is due to intrusions of aged dust particles into polluted environments which reduce the
numbers of smaller aerosols through increased coagulation with them. This results in lower cloud
reflectivity (albedo) and thus a net warming effect (Klingmüller et al., 2020). Furthermore, when
dust is above or below low-level clouds, the resulting effect of local heating is an increase in total
cloud cover due to enhanced temperature inversion or enhanced upward vertical motion,
respectively (Kok et al., 2023). On the other hand, when dust is present inside low-level clouds,
local heating enhances in-cloud evaporation, resulting in an overall decrease in cloud cover. Kok
et al. (2023) showed that the amount of desert dust in the atmosphere has increased since the mid-
19$^{th}$ century, causing an overall cooling effect on the Earth that masks up to 8% of the warming
caused by greenhouse gases. If the increase in dust were halted, the previously hidden additional
warming potential of greenhouse gases could lead to slightly faster climate warming.
$NO_3^-$ is expected to dominate the global aerosol composition in the coming decades due to the
predicted limited availability of $SO_4^{-2}$ following the abrupt decline in $SO_2$ emissions, which will
not necessarily be accompanied by proportional reductions in $NO_x$ and $NH_3$ emissions (Bellouin
et al., 2011; Hauglustaine et al., 2014). Excess $NO_3^-$ is expected to exert a cooling $RE_{ari}$ by
scattering SW radiation (Bauer et al., 2007a; Xu and Penner, 2012; Myhre et al., 2013; IPCC,
2013; Li et al., 2015), but the $RE_{aci}$ is much more complex and complicated and can lead to both
cooling and warming. Mineral dust thus becomes a key factor, as it is one of the main promoters
of $NO_3^-$ aerosol formation, providing a very suitable surface for gaseous $HNO_3$ condensation to
the aerosol phase (Karydis et al., 2011; Trump et al., 2015). In addition to $HNO_3$ adsorption,
heterogeneous reactions on the surface of dust particles are known to promote nitrate formation
(Krueger et al.,2004; Hodzic et al.,2006). The most important pathway through which this occurs
is $N_2O_5$ hydrolysis with a yield for aerosol nitrate of ~2 (Seisel et al.,2005; Tang et al.,2012). At
the same time, other reactions, such as $NO_2$ oxidation, contribute to much slower nitrate production
and are of major importance mainly during short periods of dust pollution events (Li et al., 2024).
These processes  affect not only the optical properties of dust aerosols, which will influence their
overall $RE_{ari}$, but also how they can alter cloud formation and microphysics. $NO_3^-$ aerosols increase
the hygroscopicity of mineral dust (Kelly et al., 2007) by providing layers of soluble material on
their surface, thus increasing their ability to act as CCN (Karydis et al., 2017). In doing so, they
also increase the size of dust particles through hygroscopic growth and therefore their coagulation
efficiency. Thus, nitrate-dust interactions are a complex mechanism that ultimately affects
climatology in a variety of ways. The role of mineral dust in modifying the influence of $NO_3^-$
aerosols in the global $RE_{aci}$ is not yet well understood. This study aims to focus on the extent of
the $RE_{ari}$ and $RE_{aci}$ of $NO_3^-$ aerosols and on how interactions with mineral dust regulate both on a
global scale.
This study is organized as follows: in Section 2, details of the modeling setup for conducting
the global simulations as well as the treatment of dust-nitrate interactions in the model are
discussed and the methodology for calculating the global $RE_{ari}$ and $RE_{aci}$ of $NO_3^-$ aerosols is
explained. Section 3 presents the main results for the global $RE_{ari}$ for coarse and fine $NO_3^-$ aerosols
for the base case simulation and the sensitivity cases listed in Table 1. Section 4 presents the results
for the global $RE_{aci}$ of total $NO_3^-$ aerosols, while section 5 includes the feedback mechanism of
dust-nitrate interactions with cloud microphysics. Finally, the main conclusions and a general
discussion on the scope of the study are presented in section 6.

## 2.    Methodology

### 2.1    Model Setup

The simulations were performed with the global atmospheric chemistry and climate model
EMAC (ECHAM/MESSy) (Jockel et al., 2006), which includes several submodels describing
atmospheric processes and their interactions with oceans, land, and human influences. These
submodels are linked through the Modular Earth Submodel System (MESSy) (Jockel et al., 2005)
to a base model, the 5[th] Generation European Center Hamburg General Circulation Model
(ECHAM) (Roeckner et al., 2006). The submodel system used in this work includes the MECCA
submodel, which performs the gas phase chemistry calculations (Sander et al., 2019). The SCAV
submodel is responsible for the in-cloud liquid-phase chemistry and wet deposition processes (Tost
et al., 2006; Tost et al., 2007b), while DRYDEP and SEDI are used to compute the dry deposition
of gases and aerosols and gravitational settling, respectively (Kerkweg et al., 2006). All aerosol
microphysical processes are calculated by the GMXe submodel (Pringle et al., 2010a; Pringle et
al., 2010b), where aerosols are divided into 4 lognormal size modes (nucleation, Aitken,
accumulation and coarse). Each mode is defined in terms of aerosol number concentration, number
mean dry radius, and geometric standard deviation (sigma). The mean dry radius for each mode is
allowed to vary within fixed bounds (0.5 nm – 6 nm for nucleation, 6 nm - 60 nm for Aitken, 60
nm - 700 nm for accumulation, and above 700 for coarse) and the sigma is fixed and equal to 1.59
for the first three size modes and 2 for the coarse mode.  The coagulation of aerosols is also handled
by GMXe, following Vignati et al. (2004) and the coagulation coefficients for Brownian motion
are calculated according to Fuchs and Davies (1964). The partitioning between the gas and aerosol
phases is calculated using the ISORROPIA-lite thermodynamic module (Kakavas et al., 2022) as
implemented in EMAC by Milousis et al. (2024). The optical properties of the aerosols and the
radiative transfer calculations are simulated by the submodels AEROPT (Dietmuller et al., 2016)
and RAD (Dietmuller et al., 2016), respectively. AEROPT can be called several times within a
model time step with different settings for the aerosol properties. More details are given in section
2.3.1. All cloud properties and microphysical processes are simulated by the CLOUD submodel
(Roeckner et al., 2006) using the two-moment microphysical scheme of Lohmann and Ferrachat
(2010) for liquid and ice clouds. The activation processes of liquid cloud droplets and ice crystals
follow the physical treatment of Morales and Nenes (2014) and Barahona and Nenes (2009),
respectively, as described by Karydis et al. (2017) and Bacer et al. (2018). More details are given
in Section 2.3.2.
The meteorology for each of the simulations was nudged by ERA5 reanalysis data (C3S, 2017),
thus this study estimates the radiative effect of nitrate aerosols with respect to $RE_{ari}$ and $RE_{aci}$
separately, rather than the effective (total) radiative effect, as this would require multiple free-run
simulations with prescribed sea surface temperatures for each case separately. The spectral
resolution used for each simulation was T63L31, which corresponds to a grid resolution of
$1.875° \times 1.875°$ and 31 vertical layers up to 25 km in height. The period covered by the
simulations is from 2007 to 2018, with the first year representing the model spin-up period.
Anthropogenic aerosol and trace gas emissions were taken from the CMIP6 database (O'Neill
et al., 2016) according to the SSP370 scenario. Natural $NH_3$ emissions (from land and ocean) were
based on the GEIA database (Bouwman et al., 1997), and natural volcanic $SO_2$ emissions were
taken from the AEROCOM database (Dentener et al., 2006). Biogenic NO emissions from soils
were calculated online according to the algorithm of Yienger and Levy (1995), while lightning-
produced $NO_x$ was also calculated online by the LNOx submodel (Tost et al., 2007a) using the
parameterization of Grewe et al. (2001). DMS emissions from the oceans are calculated online by
the AIRSEA submodel (Pozzer et al., 2006). Sea salt emissions are based on the AEROCOM
database (Dentener et al., 2006) following the chemical composition reported by Seinfeld and
Pandis (2016), i.e. 30.6% $Na^+$, 3.7% $Mg^+$, 1.2% $Ca^{2+}$, 1.1% $K^+$, and 55% $Cl^-$. Dust emissions are
calculated online using the parameterization of Astitha et al. (2012). In this scheme, while the
surface friction velocity is the most important parameter for the amount of the emitted dust flux,
the meteorological information for each grid cell is also taken into account. Dust particles are
emitted in the accumulation and coarse size modes of the insoluble fraction but can be transferred
to the soluble fraction after either coagulation with other soluble species and/or by condensation
of soluble material on their surface. Both processes are treated and calculated by GMXe and
ISORROPIA-lite. The emissions of mineral ions ($Ca^{2+}$, $Mg^{2+}$, $K^+$, and $Na^+$) are estimated as a
fraction of the total dust emission flux based on the soil chemical composition of each grid cell.
This is done using desert soil composition maps from Klingmüller et al. (2018) which are based
on the mineral ion fractions from Karydis et al. (2016). These mineral ions are treated as individual
species that are part of the aerosol in each size mode and are assumed to be well mixed with the
rest of the aerosol species considered (i.e., dust, black carbon, organics, inorganic ions). The
aerosol composition within each of the seven modes considered is uniform in size (internally
mixed), but may vary between modes (externally mixed).
To assess the impact of changes in mineral dust chemistry and emissions on the global $NO_3^-$
aerosol $RE_{ari}$ and $RE_{aci}$, four additional sensitivity simulations were performed (Table 1). In the
first sensitivity simulation, mineral dust is described only by a bulk, chemically inert species. In
this case, there is no uptake of $HNO_3$ by the dust particles due to acid-base interactions with the
non-volatile cations (NVCs), and so it remains in the gas phase. In the second sensitivity case, the
chemical composition of the mineral dust was assumed to be spatially uniform, with a percentage
distribution for bulk dust, $Na^+$, $K^+$, $Ca^{2+}$ and $Mg^{2+}$ particles assumed to be 94%, 1.2%, 1.5%, 2.4%
and 0.9% respectively according to Sposito (1989). Finally, two additional simulations were
performed to assess the impact of the global mineral dust budget on the results, where the dust
emission fluxes were first halved and then increased by 50% to account for the historical increase
in global dust mass load since pre-industrial times, as reconstructed by Kok et al. (2023). The
particle size distribution of the emitted dust mass remained unchanged in all sensitivity
simulations.
Overall, the EMAC model is well established in the literature for its ability to accurately predict
organic and inorganic aerosol concentrations and compositions, aerosol optical depth, acid
deposition, gas-phase mixing ratios, cloud properties, and meteorological parameters (de Meij et
al., 2012; Pozzer et al., 2012, 2022; Tsimpidi et al., 2016, 2017; Karydis et al., 2016, 2017; Bacer
et al., 2018; Milousis et al., 2024), factually replicate dust emissions (Astitha et al., 2012;
Abdelkader et al., 2015; Klingmüller et al., 2018), and provide realistic estimates for CCN and
CDNC (Chang et al., 2017; Karydis et al., 2017; Fanourgakis et al., 2019). Here, a comparison of
the performance of the model in estimating the surface mass concentrations of $PM_{2.5}$ $NO_3^-$ and
total $PM_{10}$ aerosols is provided in the supplemental material (Figures S2, S3 and Tables S1, S2).
In addition, the ability of the model to estimate CDNCs is evaluated (Figure S4 and Table S3).
The comparison is made with observations of $PM_{2.5}$ nitrate aerosols from regional networks in the
polluted northen hemisphere covering the regions of East Asia (EANET, The Acid Deposition
Monitoring Network in East Asia), Europe (EMEP, European Monitoring and Evaluation
Programme) and the USA for urban (EPA-CASTNET, U.S. Environmental Protection Agency
Clean Air Status and Trends Network) and rural (IMPROVE, Interagency Monitoring of Protected
Visual Environments) locations. The comparison with observations of surface mass $PM_{10}$ aerosols
also covers the above mentioned monitoring networks, with the exception of the EPA. Finally, the
CDNCs estimated by the base case simulation are compared with the CDNCs observed in different
regions of the planet (continental, polluted and clean marine) over different time periods, but also
altitudes, as found in Karydis et al., (2017) and all relevant references therein.

**Table 1:** Differences between the base case and all sensitivity simulations performed.

| Simulation Name | Conditions Applied |
|---|---|
| Base Case | Mineral dust ion composition according to Karydis et al. (2016)[1] |
| Sensitivity 1: Chemically Inert Dust | Mineral dust emitted exclusively as a chemically inert bulk particle |
| Sensitivity 2: Homogeneous Ion Composition | Global homogeneous ionic composition of mineral dust particles according to Sposito (1989)[2] |
| Sensitivity 3: Half Dust Scenario | 50% reduced dust emission flux |
| Sensitivity 4: Increased Dust Scenario | 50% increased dust emission flux |


## 2.2    Treatment of Dust-Nitrate Interactions

The interactions between mineral dust and nitrate aerosols play a crucial role in altering the size
distribution and optical properties of both species and can also strongly influence cloud
microphysical processes (Fig. 1). Therefore, these interactions affect both the $RE_{ari}$ and the $RE_{aci}$
of both nitrate and dust aerosols. First, the adsorption of $HNO_3$ onto the surface of dust particles
is a process that strongly promotes the formation of nitrate aerosols on dust (Karydis et al., 2016).
We treat this condensation process using the GMXe submodel. Specifically, the amount of gas
phase species that kinetically condenses within a model time step (equal to 10 minutes in this
study) is calculated according to the diffusion-limited condensation theory of Vignati et al. (2004).
The diffusive flux of gas on a single particle surface for each size mode $i$ is described by the

---

[1] The ionic composition of the dust particles with respect to the mineral ions $Ca^{2+}$, $Mg^{2+}$, $K^+$, and $Na^+$ depends on the chemical composition of the soil in each grid cell, which is estimated from the desert soil composition maps of Klingmüller et al. (2018) based on the fraction of mineral ions present found in Karydis et al. (2016).

[2] The ionic composition of the dust particles is homogeneous and held constant in all grid cells where dust is present. The dust particles are a mixture of bulk species and the mineral ions $Na^+$, $K^+$, $Ca^{2+}$ and $Mg^{2+}$ with mass fraction of 94%, 1.2%, 1.5%, 2.4% and 0.9% respectively.

condensation coefficient $C_i$ according to Fuchs and Davies (1964) and is estimated from the
following function as found in Vignati et al. (2004).
$$C_i = \frac{4\pi D r_{gi}}{\frac{4D}{svr_{gi}} + \frac{r_{gi}}{r_{gi} + \Delta}}$$

Where $r_{gi}$ is the geometric mean radius of the size mode $i$, D is the diffusion coefficient, s is an
accommodation coefficient for each gas species treated and has the assigned values of 1 for $H_2SO_4$
(Vignati et al. 2004), 0.1 for $HNO_3$, 0.064 for HCl and 0.09 for $NH_3$ (Pringle et al., 2010a; Pringle
et al., 2010b). v is the mean thermal velocity of the molecule and $\Delta$ is the mean free path length of
the gas molecule (the distance from which the kinetic regime applies with respect to the
particle).This information is then passed to the ISORROPIA-lite thermodynamic module to
calculate the gas/aerosol partitioning. Specifically, the module receives as input the ambient
temperature and humidity along with the diffusion-limited concentrations of $H_2SO_4$, $NH_3$, $HNO_3$,
and HCl, the concentrations of the non-volatile cations (NVCs) $Na^+$, $K^+$, $Ca^{2+}$ and $Mg^{2+}$, and the
concentrations of the ions $SO_4^{2-}$, $NO_3^-$, $NH_4^+$, and $Cl^-$ present in the aerosol phase from the previous
time step. The module then calculates the equilibrium reactions of the $NO_3^-$ anion with the NVCs,
depending on their abundance with respect to the $SO_4^{2-}$ anion, taking into account mass
conservation, electroneutrality, water activity equations and precalculated activity coefficients for
specific ionic pairs (Fountoukis and Nenes, 2007; Kakavas et al., 2022). Therefore, in all cases
where mineral dust is considered chemically active, all reactions of nitrate aerosols with NVC are
treated. The salts that may be formed are assumed to be completely deliquesced as follows:
$Ca(NO_3)_2 \rightarrow Ca^{2+}_{(aq)} + 2NO^-_{3(aq)}$
$NaNO_3 \rightarrow Na^+_{(aq)} + NO^-_{3(aq)}$
$KNO_3 \rightarrow K^+_{(aq)} + NO^-_{3(aq)}$
$Mg(NO_3)_2 \rightarrow Mg^{2+}_{(aq)} + 2NO^-_{3(aq)}$

Salt deliquescence over a range of relative humidities is treated by the Mutual Deliquescence
Relative Humidity (MDRH) approach of Wexler and Seinfeld (1991). In a multicomponent salt
mixture, the MDRH determines the humidity value above which all salts are considered to be
saturated. In this study, if the wet aerosol is below the MDRH, it does not crystalize and remains
in a supersaturated aqueous solution (Kakavas et al., 2022), with all salts completely deliquesced.
More information on equilibrium reactions and equilibrium constants as well as the corresponding
thermodynamic equilibrium calculations can be found in Fountoukis and Nenes (2007). It should
be noted that in this study nitrate production on dust particles does not only occur via the
thermodynamic equilibrium between gas-phase $HNO_3$ and particulate nitrate, but also via
heterogeneous chemistry by hydrolysis of $N_2O_5$ on the dust surface. This chemical formation
pathway is the most dominant for heterogeneous nitrate production (Seisel et al., 2005; Tang et
al., 2012), while others, such as $NO_2$ oxidation during dust pollution events over polluted regions
(Li et al., 2024), do not show such high yields under normal conditions. On the other hand,
consideration of sulphate production by heterogeneous chemistry on dust would theoretically
result in slightly reduced amounts of particulate nitrate in some cases due to  acidification of dust
particles inhibiting partitioning of HNO₃ to the aerosol phase (Nenes et al., 2020). Overall, full
consideration of heterogeneous chemistry on dust could change simulated nitrate aerosol
concentrations only slightly and episodically, and therefore changes to radiative effect estimates
are not expected to be critical.
The coating of dust particles by nitrate aerosols during gas/aerosol partitioning calculations is
an important process that leads to an increase in dust solubility and hygroscopicity (Laskin et al.,
2005). Therefore, after these processes have taken place, a large fraction of the originally insoluble
dust particles has become soluble (Fig. 1a), which leads to changes in their optical properties, as
their increased ability to absorb water makes them more efficient in extinguishing SW radiation
and absorbing and emitting LW radiation (Fig. 1a, 1b) (Kok et al., 2023). The transfer to the soluble
fraction after coating with soluble material is handled by the GMXe submodel, which also provides
key aerosol attributes necessary for the calculation of the dust optical properties (see Section 2.3).

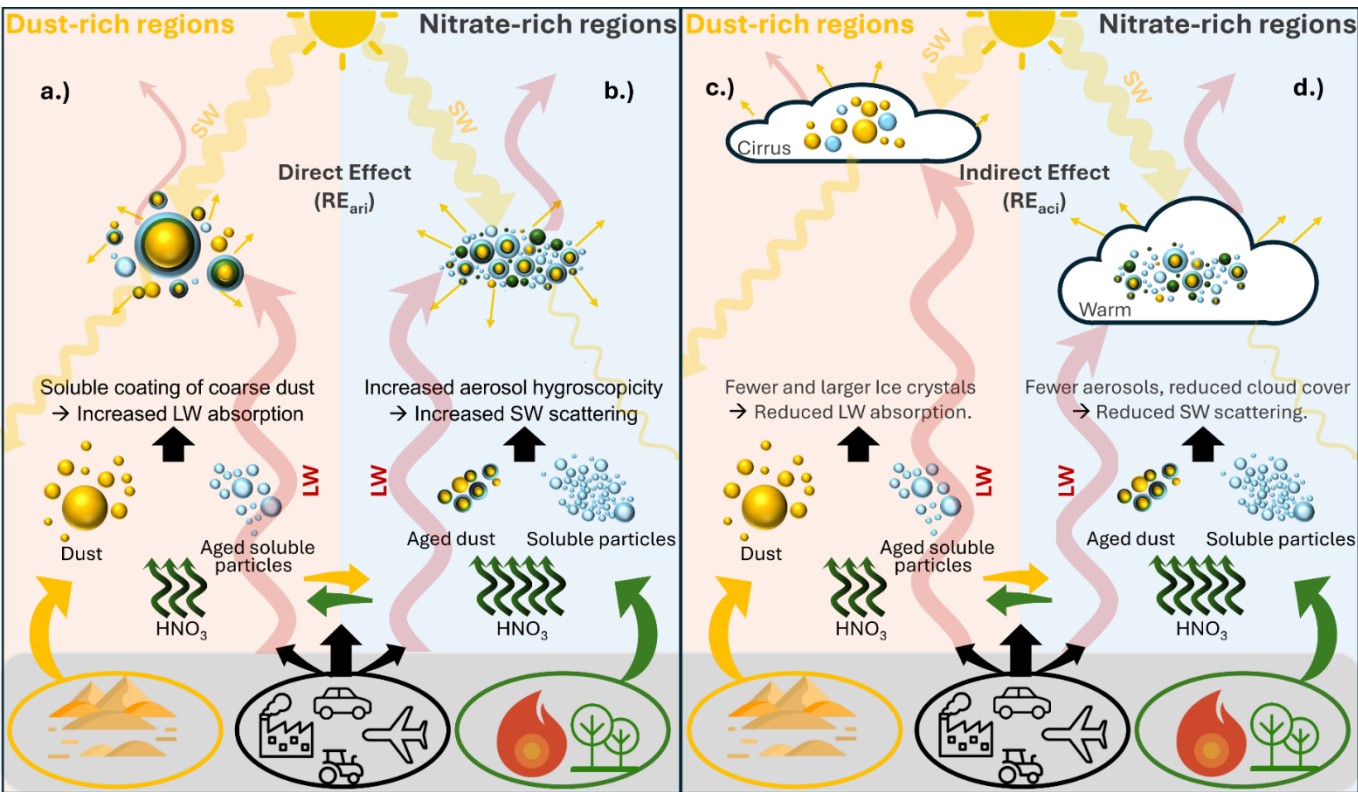

**Figure 1:** Conceptual illustration of how dust-nitrate interactions affect the total $NO_3^-$ (left) $RE_{ari}$ and
(right) $RE_{aci}$. **a)** In dust-rich environments, nitric acid transported from anthropogenic pollution and biomass
burning regions interacts with mineral cations to form a soluble coating on the surface of dust particles. The
dominant effect of these interactions is an enhanced LW absorption (warming $RE_{ari}$) by the coarse dust
particles. **b.)** In nitrate-rich environments, the intrusion of dust particles and their subsequent interaction
with freshly formed nitric acid leads to an overall increase in aerosol hygroscopicity and thus a stronger
SW reflection (cooling $RE_{ari}$). **c.)** In dust-rich environments, the number of ice crystals in cirrus clouds is
reduced while their size is increased due to the interaction of dust particles with the transported $HNO_3$. This
results in an optical thinning of the ice clouds, which leads to less trapping of outgoing LW radiation
(cooling $RE_{aci}$). **d.)** In nitrate-rich environments, the increased wet radius of aged dust particles leads to
enhanced coagulation with smaller particles, resulting in a decrease in the number of smaller aerosols and,
in turn, a decrease in the number of activated particles in cloud droplets by smaller aerosols, which
ultimately leads to a reduction in the backscattering of SW radiation by warm clouds (warming $RE_{ari}$).

In general, the changes in the properties of dust particles through their interactions with nitrate
aerosols will result in more efficient removal rates, mainly through wet deposition, due to their
higher hygroscopicity and increased size (Fan et al., 2004). The reduced number of dust particles
that can act as ice nuclei (IN) and their increased size can lead to an optical thinning of cirrus
clouds (Fig. 1c) (Kok et al., 2023). Furthermore, the changes induced by dust-nitrate interactions
reduce the activation of smaller aerosols in warm clouds (Fig. 1d). In particular, the enhanced
hygroscopicity of dust particles will lead to a faster depletion of the available supersaturation, as
they act as giant CCN that absorb large amounts of water vapor to activate into cloud droplets
(Karydis et al., 2017). In addition, the population of smaller aerosols will also be depleted by
increased coagulation with the large dust particles. As a consequence of the different degrees of
complexity of the dust-nitrate interactions, it is very important to note that they do not always
result in a linear response in terms of how they affect climate through their subsequent interactions
with radiation, clouds, or both.

## 2.3    Radiative Effect Calculation

To calculate the global $RE_{ari}$ and $RE_{aci}$ of $NO_3^-$ aerosols, the optical properties from the
AEROPT submodel and the radiative transfer calculations from the RAD submodel were used.
First, AEROPT provides the aerosol extinction (absorption and scattering) coefficients, the single
scattering albedo, and the aerosol asymmetry factor for each grid cell with a vertical distribution
analogous to the vertical resolution used. The GMXe submodel is used to provide input of aerosol
attributes for the calculation of aerosol optical properties, which is done online using 3D look-up
tables. The tables provide information on the real and imaginary parts of the refractive index and
the Mie size parameter per size mode (Dietmuller et al., 2016). Then, the radiative scheme of RAD
uses the particle number weighted average of the extinction cross section, the single scattering
albedo, and the asymmetry factor as input for the radiative transfer calculations. In addition to
AEROPT, RAD takes input from the submodels ORBIT (Earth orbital parameters), CLOUDOPT
(cloud optical properties) (Dietmuller et al., 2016), and IMPORT (import of external datasets) to
calculate the radiative transfer properties for longwave and shortwave radiation fluxes separately.
Both the AEROPT and RAD submodels can be invoked multiple times within a model time step,
each time with different settings for the aerosol optical properties, allowing radiative transfer
estimates for identical climatological conditions. This is of paramount importance for the
calculation of the $RE_{ari}$ of aerosols since any effects due to possibly different climatological
conditions must be eliminated. Henceforth, all references to RE estimates, as well as net,
longwave, and shortwave flux quantities, will refer to the top of the atmosphere (TOA) only.

### 2.3.1  Radiative Effect from Aerosol-Radiation Interactions (REari)

To estimate the global $RE_{ari}$ of all aerosols as well as that of total, coarse, and fine $NO_3^-$ aerosols,
3 simulations were performed for each sensitivity case in Table 1. In the first simulation all aerosol
species are present. In the second simulation $NO_3^-$ aerosols are completely removed by turning off
their formation by removing the pathway of $HNO_3$ formation through both $NO_2$ oxidation and
N₂O₅ hydrolysis, leaving no available HNO₃ to condense on the aerosol via equilibrium
partitioning and form nitrate. In the third simulation, coarse mode $NO_3^-$ aerosols are removed by
allowing HNO₃ to condense only on the fine mode (i.e., the sum of the three smaller lognormal
size modes: nucleation, Aitken, and accumulation). For each of these three simulations, the
radiative transfer routines are called twice for each time step. One call uses the normal aerosol
optical properties of the existing population, and the other call uses an aerosol optical depth equal
to 0 to emulate an atmosphere without aerosols. Essentially, the global $RE_{ari}$ of each simulation
can be calculated by taking the difference between the net fluxes between the two calls. More
specifically, the first simulation will yield the $RE_{ari}$ of the total aerosol load ($F_{1,ari}$ hereafter), the
second simulation will yield the $RE_{ari}$ of all aerosols except $NO_3^-$ ($F_{2,ari}$ below), and the third
simulation will yield the $RE_{ari}$ of all aerosols except the coarse mode $NO_3^-$ ($F_{3,ari}$ below). Since the
above estimates of the radiative effect were computed using the exact same climatology, its effect
was effectively eliminated. However, in order to isolate the $NO_3^-$ aerosol radiative effect, it is also
essential to disable any aerosol-cloud interactions, otherwise the cooling effect would be severely
underestimated because cloud scattering would make aerosol scattering less relevant (Ghan et al.,
2012). For this purpose, the simplest cloud scheme available in the EMAC model is used, which
calculates the cloud microphysics according to Lohmann and Roeckner (1996) who empirically
relate the cloud droplet number concentration to the sulfate aerosol mass (Boucher and Lohmann
1995) and specifically to its monthly mean values as derived from the sulfur cycle of the ECHAM5
circulation model (Feichter et al., 1996). The cloud coverage is estimated according to Tompkins
(2002) with the use of prognostic equations for the water phases and the distribution moments. To
disable aerosol-cloud interactions, no aerosol activation routines are used to avoid coupling with
the activation schemes. Overall, the global $RE_{ari}$ of total, coarse, and fine $NO_3^-$ aerosols are
obtained as follows:
●    $F_{NO3,ari}(F_{N,ari}) = F_{1,ari} - F_{2,ari}$
●    $F_{coarseNO3,ari}(F_{cN,ari}) = F_{1,ari} - F_{3,ari}$
●    $F_{fineNO3,ari}(F_{fN,ari}) = F_{3,ari} - F_{2,ari}$

### 382   2.3.2   Radiative Effect from Aerosol-Cloud Interactions ($RE_{aci}$)

In this work we estimate the effect of total $NO_3^-$ aerosols on the calculated global $RE_{aci}$.
Climatology plays a crucial role in aerosol-cloud interactions and simulating a "fine-only $NO_3^-$
atmosphere", as done for the $RE_{ari}$ calculations, would produce an unrealistic climatological
scenario, since coarse-mode $NO_3^-$ is strongly associated with cations in mineral dust particles
(Karydis et al., 2016), making them quite effective as CCN (Karydis et al., 2017). Therefore, the
$RE_{aci}$ calculations require 2 additional simulations for each sensitivity case separately: one with all
aerosols present and one with the entire $NO_3^-$ aerosol load removed by turning off their formation
as described in the previous section. The global $RE_{aci}$ is then given by:
●    $F_{NO3,aci}(F_{N,aci}) = FF_N - F_{N,ari}$
where $FF_N$ is the total $NO_3^-$ aerosol feedback radiative effect. Since $F_{N,ari}$ is calculated using the
methodology described in Section 2.3.1, it is only necessary to estimate $FF_N$. This is equal to the
difference in net fluxes between the two additional simulations. There is no need to emulate an
aerosol-free atmosphere here since any differences induced by different climatologies must be
included. The two simulations performed for the calculation of $FF_N$ use the cloud formation
scheme as described in Lohmann and Ferrachat (2010), which uses prognostic equations for the
water phases and the bulk cloud microphysics. In addition, the empirical cloud cover scheme of
Sundqvist et al. (1989) is used. For aerosol activation, the CDNC activation scheme of Morales
and Nenes (2014) is used, which includes the adsorption activation of mineral dust as described in
Karydis et al. (2017). The effect of dust-nitrate interactions on clouds presented here refers to the
lowest level of cloud formation at 940 hPa. For the ICNC activation, the scheme of Barahona and
Nenes (2009) is used, which calculates the ice crystal size distribution through heterogeneous and
homogeneous freezing as well as ice crystal growth.

# 3.  Radiative Effect from Aerosol-Radiation Interactions ($RE_{ari}$)

## 3.1  Base Case

The global average $RE_{ari}$ of total $NO_3^-$ aerosols at the top of the atmosphere was found to be -
0.11 W/m$^2$, which is within the reported range of the estimated present day all-sky direct radiative
effect of total $NO_3^-$ aerosols by other studies (Liao et al., 2004; Bauer et al., 2007a; Bauer et al.,
2007b; Bellouin et al., 2011; Xu and Penner, 2012; Heald et al., 2014) (Table S4). The $NO_3^-$
cooling of the $RE_{ari}$ calculated by EMAC is driven by the scattering of SW radiation (equal to -
0.34 W/m$^2$), which outweighs the warming due to absorption of LW radiation (equal to +0.23
W/m$^2$) (Table 2). The $RE_{ari}$ of the total $NO_3^-$ aerosol shows a clearly contrasting behavior with
respect to the size mode considered (Table 2; Figure 2).
In particular, the coarse particles show a net warming effect of +0.17 W/m$^2$ (Fig. 2i) and
contribute to 96% of the LW warming of the total nitrate, while only contributing 15% of the
radiative cooling in the SW spectrum (-0.05 W/m$^2$). The LW warming is strongest over the dust
belt zone and especially over the Sahara, the Middle East and the northern face of the Himalayan
plateau, while the contribution over other arid regions such as the Atacama, Gobi, Taklimakan and
Mojave deserts is significant. These regions are characterized by moderate to high concentrations
of coarse $NO_3^-$ aerosols due to the adsorption of $HNO_3$ on desert soil particles (Karydis et al., 2016;
Milousis et al., 2024). Therefore, the warming due to absorption of terrestrial LW radiation by
coarse-mode nitrates interacting with mineral dust is the strongest over these areas (see Fig. 1a),
ranging from +1.5 W/m$^2$ to +5 W/m$^2$ (Fig. 2iii). On the other hand, the cooling exerted by coarse
nitrate aerosol through the SW $RE_{ari}$ is more pronounced over areas where it interacts strongly
with high concentrations of mineral dust particles (see Fig. 1b). Such areas include the Congo
Basin, where $HNO_3$ from tropical forest biomass burning interacts with Saharan mineral dust
particles; the Middle East and North Indian regions, where anthropogenic $HNO_3$ emissions interact
with mineral dust particles from the Sahara and Taklimakan deserts, respectively; and the East
Asian region, where $HNO_3$ emissions from Chinese megacities interact with mineral dust particles
from the Gobi Desert. These regions can lead to an average cooling of up to -3.5 W/m$^2$ (Fig. 2v).
Interestingly, there is no significant cooling from SW interactions over the Sahara for the coarse
mode. This phenomenon can be attributed to two factors, the first related to nitrate-dust
interactions and the second related to the characteristics of the region. Specifically, because the
underlying desert surface is very bright, its absorption in this part of the spectrum is less than that
of the particles above it, which means that the desert surface can scatter radiation more effectively
than the particles above it. This is further enhanced by the growth of coarse mode particles there
(see Fig. 4x and section 5.1) which increases the absorption cross section of the particles. All this
leads to an overall attenuation of the cooling effect over this region and sometimes even to local
warming (Fig. 2v).
441        In contrast to the radiative effect of coarse $NO_3^-$ particles, the $RE_{ari}$ of fine $NO_3^-$ particles is an
overall cooling of -0.28 W/m$^2$ (Fig. 2ii). Fine nitrates have a negligible 4 % contribution to the
warming in the LW spectrum (Fig. 2iv) but account for 85 % of the net cooling of the total nitrate
aerosols (Fig. 2vi). The cooling induced by fine $NO_3^-$ aerosols from scattering of SW radiation is
stronger (up to -5 W/m$^2$) over regions of high anthropogenic activity, particularly the East Asian
and Indian regions, where fine nitrates dominate the total nitrate aerosol load. The regions of West
Africa and the Amazon Basin are characterized by moderate fine nitrate concentrations, and the
cooling observed there is enhanced by $HNO_3$ associated with biomass burning interacting with
fresh and aged Saharan dust particles, respectively, which are dominated by accumulation mode
sizes in the absence of coarse mode nitrates. Finally, other polluted regions such as North America
and Europe also show SW cooling up to -2 W/m$^2$.

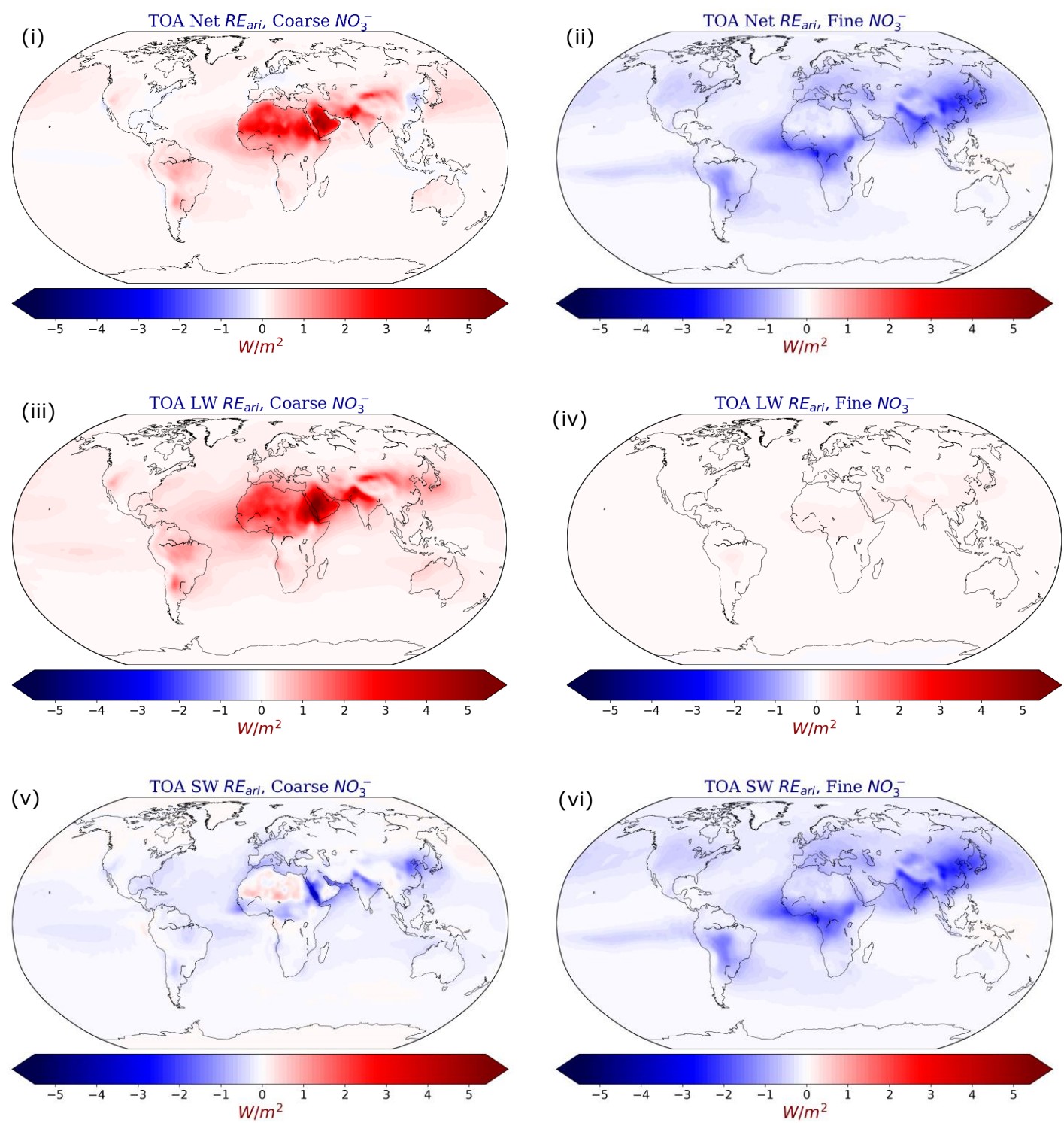


**Figure 2:** Global mean TOA net $RE_{ari}$ for (i) coarse and (ii) fine NO3- aerosols; longwave $RE_{ari}$ for (iii)
coarse and (iv) fine NO3- aerosols; shortwave $RE_{ari}$ for (v) coarse and (vi) fine NO3- aerosols, as calculated
by EMAC from the base case simulation.
**Table 2:** Net, longwave, and shortwave global mean TOA $RE_{ari}$ of total, coarse, and fine $NO_3^-$
aerosols for the base case and each sensitivity case simulations.

| Simulation | Aerosol Component | TOA $RE_{ari}$ (W/m$^2$) | | |
|---|---|---|---|---|
| | | **Net** | **LW** | **SW** |
| **Base Case** | Total $NO_3^-$ | - 0.11 | + 0.23 | - 0.34 |
| | Coarse $NO_3^-$ | + 0.17 | + 0.22 | - 0.05 |
| | Fine $NO_3^-$ | - 0.28 | +0.01 | - 0.29 |
| **Chemically Inert Dust** | Total $NO_3^-$ | - 0.09 | + 0.11 | - 0.20 |
| | Coarse $NO_3^-$ | + 0.07 | + 0.10 | - 0.03 |
| | Fine $NO_3^-$ | - 0.16 | + 0.01 | - 0.17 |
| **Homogeneous Ion Composition** | Total $NO_3^-$ | - 0.09 | + 0.18 | - 0.27 |
| | Coarse $NO_3^-$ | + 0.13 | + 0.17 | - 0.04 |
| | Fine $NO_3^-$ | - 0.22 | + 0.01 | - 0.23 |
| **Half Dust Scenario** | Total $NO_3^-$ | - 0.08 | + 0.19 | - 0.27 |
| | Coarse $NO_3^-$ | + 0.15 | + 0.18 | - 0.03 |
| | Fine $NO_3^-$ | - 0.23 | + 0.01 | - 0.24 |
| **Increased Dust Scenario** | Total $NO_3^-$ | - 0.10 | + 0.27 | - 0.37 |
| | Coarse $NO_3^-$ | + 0.20 | + 0.26 | - 0.06 |
| | Fine $NO_3^-$ | - 0.30 | + 0.01 | - 0.31 |


## 3.2    Sensitivity of RE$_{ari}$ Estimates

The comparison of the calculated total $NO_3^-$ radiative effect due to interactions with net, LW, and SW radiation for the sensitivity cases listed in Table 1 can be found in Table 2, which shows each of the estimates. Consideration of nitrate interactions with mineral dust cations can greatly affect the $NO_3^-$ RE$_{ari}$ estimates. Assuming that mineral dust particles are inert, the estimated warming due to LW radiation interactions for total nitrate aerosols is 52% weaker than in the base case where dust reactivity is considered. Similarly, the cooling effect exerted by all nitrate aerosols through interactions with SW radiation is estimated to be 41% weaker under the assumption that mineral dust is non-reactive. Both estimates are lower when mineral dust is assumed to be chemically inert, since $HNO_3$ is no longer effectively adsorbed on dust particles. However, since both the estimated warming and cooling are weaker, the effects partially cancel each other out, resulting in a net cooling effect (-0.09 W/m$^2$) that is 18% weaker compared to the base case calculations. Assuming a homogeneous ionic composition for the dust, results in SW cooling and LW warming for total nitrate aerosols being 21% and 22% lower, respectively, weakening the estimate for the net cooling RE$_{ari}$ by 18% (-0.09 W/m$^2$). The net direct radiative effect of total $NO_3^-$ is the same for the cases where dust is assumed to have a homogeneous chemical composition and where it has no chemical identity, indicating the importance of both aspects for the impact of dust-nitrate interactions on the direct radiative effect.

In the Half Dust scenario, the total nitrate aerosol LW warming estimate is 17% weaker than in the base case, while the total nitrate aerosol SW estimate is even more so (21%), resulting in a lower net cooling estimate of -0.08 W/m$^2$. Finally, the Increased Dust scenario shows the strongest total nitrate aerosol LW warming effect (17% increase over the base case) due to an increase in coarse mode nitrate. At the same time, the cooling effect of total nitrate aerosols due to interactions with SW radiation shows a smaller increase of 9%. Thus, accounting for the historical increase in mineral dust emissions results in a net cooling estimate of -0.10 W/m$^2$, which is smaller than the base case. Interestingly, the behavior of the global total $NO_3^-$ RE$_{ari}$ does not exhibit linearity with respect to the global dust load. This is not surprising since the nitrate-dust interactions themselves are not linearly correlated, and a given increase or decrease in dust emissions does not lead to an analogous change in nitrate aerosol levels. For example, Karydis et al. (2016) have shown that moving from a scenario in which nitrate-dust chemistry is not considered to one in which it is, but with half dust emissions, resulted in a 39% increase in the tropospheric burden of nitrate aerosols. However, moving from a scenario with half to full dust emissions, the corresponding increase was only 9%. In our case, moving from the chemically inert dust scenario to the half dust scenario led to an 18% increase in atmospheric nitrate aerosol burden, while moving from the half dust scenario to the base case led to an additional 8% increase, and finally moving from the base case to the increased dust scenario led to an even smaller increase of 5%.

There are several reasons for this non-linearity between changes in dust load and nitrate production. Firstly, since the adsorption of $HNO_3$ onto dust particles is the main driver of nitrate production on dust, over desert areas (where the change in dust load takes place) the amount of nitric acid present is the limiting factor for such production, rather than the amount of dust itself. Secondly, when more dust is present in the atmosphere, the combination of its increased coating with the higher aerosol numbers, tends to result in its  more efficient removal by wet deposition as

well as coagulation. This inherently affects nitrate production, which does not increase in
proportion to the increase in dust.

## 4 Radiative Effect from Aerosol-Cloud Interactions (RE$_{aci}$)

### 4.1 Base Case

The global average RE$_{aci}$ of total NO$_3^-$ aerosols at the top of the atmosphere was found to be
+0.17 W/m$^2$. In contrast, an estimate of the RE$_{aci}$ of nitrate aerosols by Xu and Penner (2012)
showed only a trivial cooling effect for particulate NO$_3^-$ (-0.01 W/m$^2$). Similar to the RE$_{ari}$, the net
RE$_{aci}$ estimated by EMAC is driven by the effect on the SW part of the spectrum, which causes a
warming effect of +0.27 W/m$^2$, while the effect on the LW radiation causes an average cooling of
-0.10 W/m$^2$ (Table 3). Overall, the net RE$_{aci}$ of total NO$_3^-$ aerosols is reversed compared to the net
RE$_{ari}$, i.e. RE$_{aci}$ exerts a strong cooling effect over regions where RE$_{ari}$ exerts a warming effect and
vice versa (Fig. 3i). The reason for this is that the regions contributing to a cooling RE$_{ari}$ are
dominated by smaller sized nitrate aerosols and vice versa. Therefore, the size characteristics of
the dominant nitrate aerosol population lead to different effects on the cloud optical properties as
discussed in section 1. For example, as the dominance of smaller nitrate aerosols decreases over a
particular region, the optical thinning of low-level clouds will have an opposite effect on the RE$_{aci}$
(Fig. 1d). Details of the mechanism by which nitrate-dust interactions affect cloud microphysical
processes are discussed in section 5. Over North America and Europe, RE$_{aci}$ causes a warming
effect of up to +3 W/m$^2$, driven solely by the effect on SW radiation (Fig. 3iii). Over the regions
of East Asia and the Amazon and Congo basins, RE$_{aci}$ reaches a maximum of +5 W/m$^2$, driven by
both the effect on the SW (up to +4 W/m$^2$) and LW (up to +1.5 W/m$^2$) parts of the radiation
spectrum. The cooling effect of RE$_{aci}$ (up to -2 W/m$^2$) extends mainly between the equatorial line
and the Tropic of Cancer, mainly due to the interaction of nitrate aerosols with desert dust particles
(e.g. from the Sahara) and their effect on the terrestrial spectrum (LW) (Figs. 1c & 3ii). The cooling
effect of dust interactions with anthropogenic particles in the LW spectrum corroborates the
findings of Klingmüller et al. (2020) and is attributed to the reduced ice-water path due to the
depletion of small aerosols, which in turn leads to less trapped outgoing terrestrial radiation. In
addition, Kok et al. (2023) note how the presence of dust particles leads to an optical thinning of
cirrus clouds by reducing the number of ice crystals while increasing their size, which also leads
to less trapping of outgoing LW radiation and thus a cooling effect (Fig. 1c). On the other hand,
the warming effect of dust interactions with anthropogenic particles in the SW spectrum requires
further investigation and is therefore discussed in more detail in Section 5.

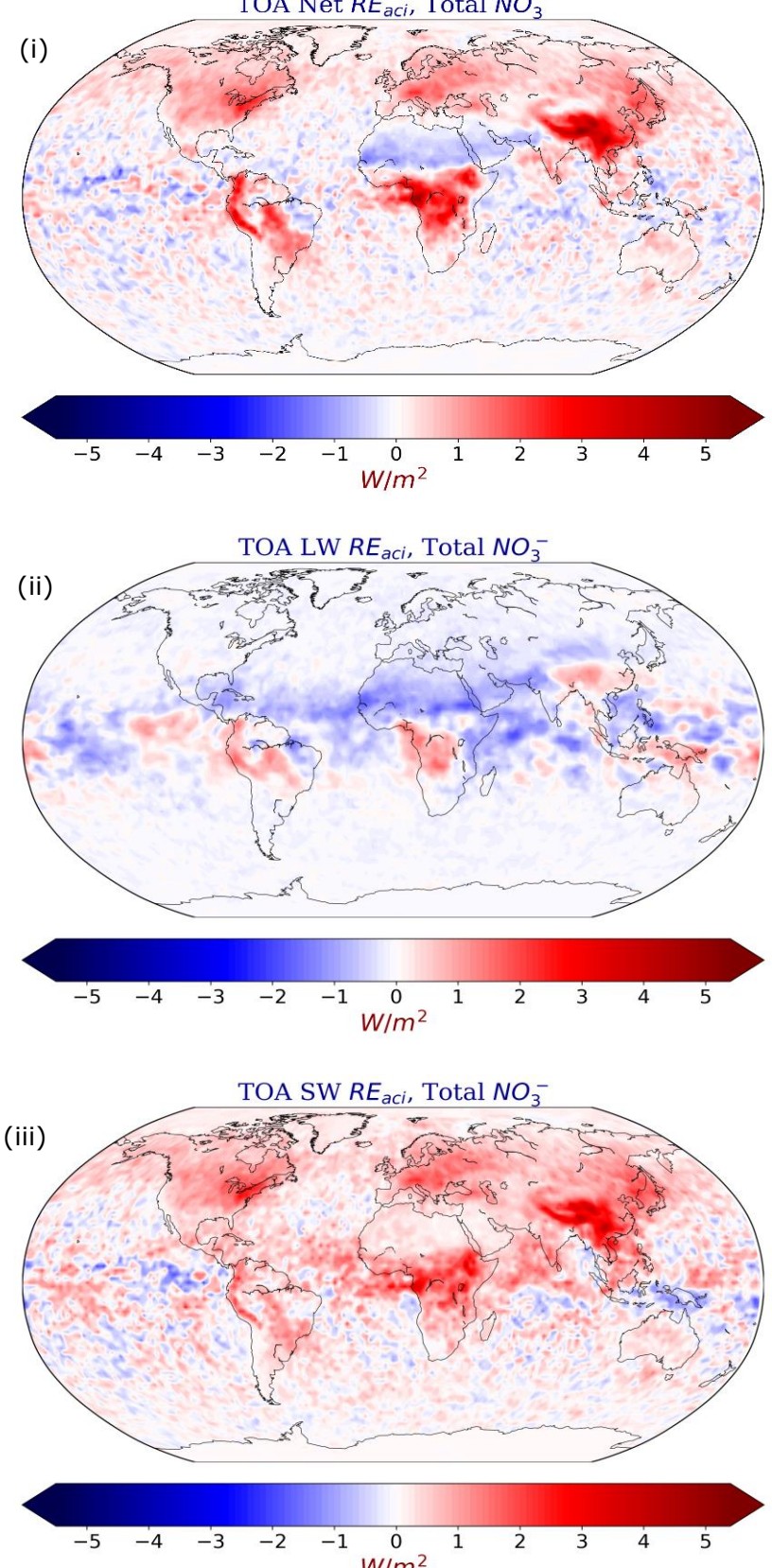

**Figure 3:** Global mean TOA RE$_{aci}$ for total NO$_3^-$ aerosols. Estimates for (i) net, (ii) longwave, and (iii) shortwave, as calculated by EMAC from the base case simulation.

**Table 3:** Net, longwave, and shortwave global mean TOA $RE_{aci}$ of total $NO_3^-$ aerosols for the base case and each sensitivity case simulations.

| Simulation | TOA $RE_{aci}$ (W/m$^2$) | | |
|:---:|:---:|:---:|:---:|
| | Net | LW | SW |
| Base Case | + 0.17 | - 0.10 | + 0.27 |
| Chemically Inert Dust | + 0.11 | - 0.06 | + 0.17 |
| Homogeneous Ion Composition | + 0.13 | - 0.09 | + 0.22 |
| Half Dust Scenario | + 0.15 | - 0.08 | + 0.23 |
| Increased Dust Scenario | + 0.14 | - 0.11 | + 0.25 |

## 4.2   Sensitivity of $RE_{aci}$ Estimates

Table 3 shows the comparison of the net, LW, and SW contributions of total $NO_3^-$ to the $RE_{aci}$ at the top of the atmosphere as calculated by the base case simulation and all sensitivity cases considered. By assuming a chemically inert dust, the calculated net $RE_{aci}$ of nitrate decreases by 35%, resulting in a net warming of +0.11 W/m$^2$. As with the $RE_{ari}$ estimate, this sensitivity case produces the largest deviation from the base case among all sensitivity simulations, for both the SW (37% less warming) and LW (40% less cooling) estimates. This is due to the fact that the absence of dust-nitrate interactions does not have such a large impact on the population of both aerosols and activated particles (see also Section 5). The assumption of a homogeneous ionic composition of the mineral dust leads to a weakened LW cooling estimate of 10% and a weakened SW warming estimate of 19% resulting in a net $NO_3^-$ $RE_{aci}$ of +0.13 W/m$^2$ (24% lower than in the base case).

The reduced dust emissions result in a 15% weaker warming in the SW spectrum and a 20% weaker cooling in the LW spectrum, leading to an overall $NO_3^-$ $RE_{aci}$ of +0.15 W/m$^2$ (12% weaker than the base case scenario). This is because the reduced loading of nitrate aerosols, especially in the coarse mode, in the half dust scenario results in less absorption of LW radiation (Fig. 1c) (hence less cooling). Similarly, the effect of dust-nitrate interactions on the activation of smaller particles (Fig. 1d) is less drastic and results in a weaker inhibition of SW radiation scattering (hence less warming, see also Section 5). Finally, increased dust emissions in the increased dust scenario show a 10% increase in the LW cooling and an 8% decrease in the SW warming effect, surprisingly resulting in a net warming (+0.14 W/m$^2$) that is lower than in the half dust scenario. The reason that this scenario results in more LW cooling than the base case is that the increased amount of dust particles leads to even more optical thinning of the ice clouds, and therefore even less trapping of LW radiation (more cooling). However, the reason why the SW warming estimate is lower than the base case is more complicated. First, the transition from the half dust scenario to the base case and then to the increased dust scenario does not lead to an analogous increase in the nitrate aerosol

burden (see Section 3.2). Moreover, since the number of aerosols has increased from the increased
dust scenario to the base case, but the relative humidity has remained largely the same, there is
more competition for water vapor because it is now distributed over a larger population. As a
result, the wet radius increase in the presence of nitrates is not as strong in the increased dust
scenario compared to the base case, and the depletion of smaller sized particles is also not as strong
(not shown). The implications of the depletion of the aerosol population in the presence of nitrate
aerosols on the microphysical processes of warm clouds, and consequently on SW warming, are
discussed in the next section.

## 574 5    Effect Of $NO_3^-$ Aerosols on Cloud Microphysics

### 575 5.1    Maximum Supersaturation, Hygroscopicity and Wet Radius

To further investigate the cause of the positive $RE_{aci}$ induced by the $NO_3^-$ aerosols, their effect
on the aerosol population characteristics as well as on the cloud microphysics is investigated, with
respect to the lowest forming cloud level of 940 hPa. For this purpose, a sensitivity simulation is
performed assuming a 'nitrate aerosol free' (NAF) atmosphere, in which the formation of $NO_3^-$
aerosols has been switched off, but an advanced cloud scheme is considered which is the same as
the one described in Section 2.3.2. Essentially the same setup that was used for the estimation of
the total nitrate aerosol feedback radiative effect. This simulation is used to determine whether the
presence of $NO_3^-$ aerosols has a significant effect on the hygroscopicity and size of atmospheric
aerosols and ultimately on the maximum supersaturation developed during cloud formation. Over
polluted areas affected by transported dust air masses from surrounding arid areas, the presence of
$NO_3^-$ aerosols can increase the CCN activity of the large mineral dust particles, resulting in a
reduction of the maximum supersaturation and inhibiting the activation of the small anthropogenic
particles into cloud droplets (Klingmüller et al., 2020). Results from the NAF sensitivity
simulation support this hypothesis over parts of Eastern and Central Asia, where the maximum
supersaturation decreases by up to 0.05%. In contrast, the presence of $NO_3^-$ aerosols increases
maximum supersaturation by up to 0.2% over North America, Europe, the Middle East, and parts
of southern Asia (Fig. 4ii). Therefore, changes in maximum supersaturation caused by the presence
of $NO_3^-$ aerosols cannot explain their warming effect through the $RE_{aci}$.
The presence of $NO_3^-$ has a significant effect on the hygroscopicity of both fine and coarse
aerosols and consequently on their wet radius, as shown in Figures 1a,b & 4. This is most evident
for coarse desert dust particles, which mix with $NO_3^-$ aerosols from urban and forest regions,
increasing their hygroscopicity by an order of magnitude (up to 0.1), especially over the African-
Asian dust belt and the Atacama Desert in South America (Fig. 4vi). Aerosol hygroscopicity is
similarly increased for the fine mode particles both near arid regions and over the highly
industrialized region of Southeast Asia (Fig. 4iv). The low values of the hygroscopic parameter of
the fine aerosol population, especially over the dust belt zone, are largely due to the higher
proportion of insoluble fine particles present over these regions (Figure S5). This is also observed
over other regions with similarly low fine aerosol hygroscopicity (South Africa, South America
and Western U.S). Nevertheless, the estimates of aerosol kappa values at 940 hPa are broadly
consistent with the results of Pringle et al., (2010c). On the other hand, the aerosol hygroscopicity
for the two size modes is only slightly reduced, by up to 0.06 (or <10%) over the oceans and coasts
of Europe and East Asia, due to interactions of $NO_3^-$ with sea salt particles, reducing their

hygroscopicity. The increased ability of both coarse dust aerosols and smaller aerosols to absorb water leads to an increase in their wet radius, but in different parts of the world. For example, fine particle sizes increase by up to 0.04 μm (up to 40%) mostly over regions of high anthropogenic activity (North America, Europe, and East Asia) (Fig. 4viii). On the other hand, coarse mode particle sizes are increased by up to 0.1 μm (up to 10%) over the forests of central Africa and the African-Asian dust belt zone (Fig. 4x), while showing a similar decrease near the coasts of the polluted northern hemisphere due to the effect of $NO_3^-$ on the hygroscopicity of sea salt.

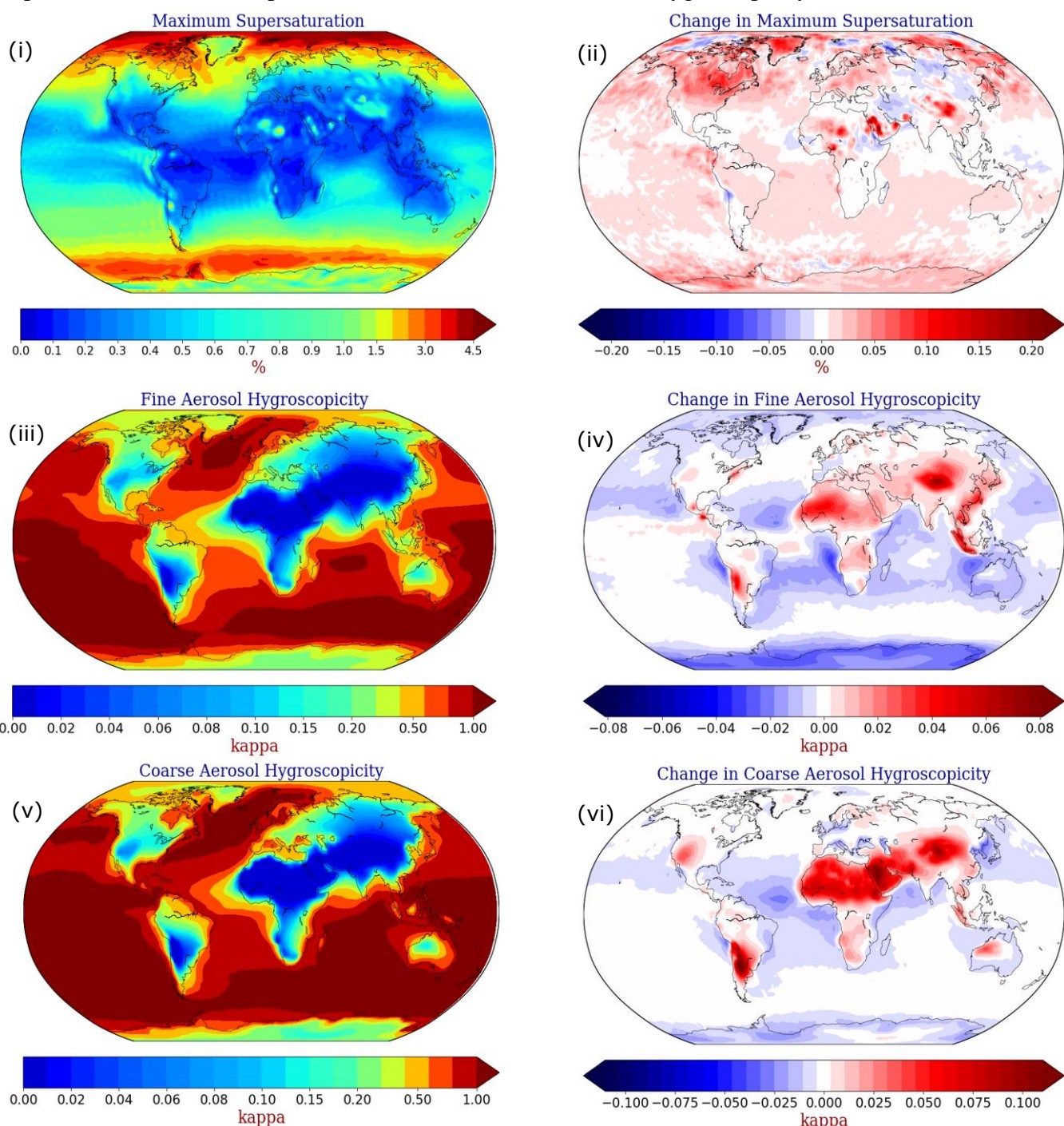

615

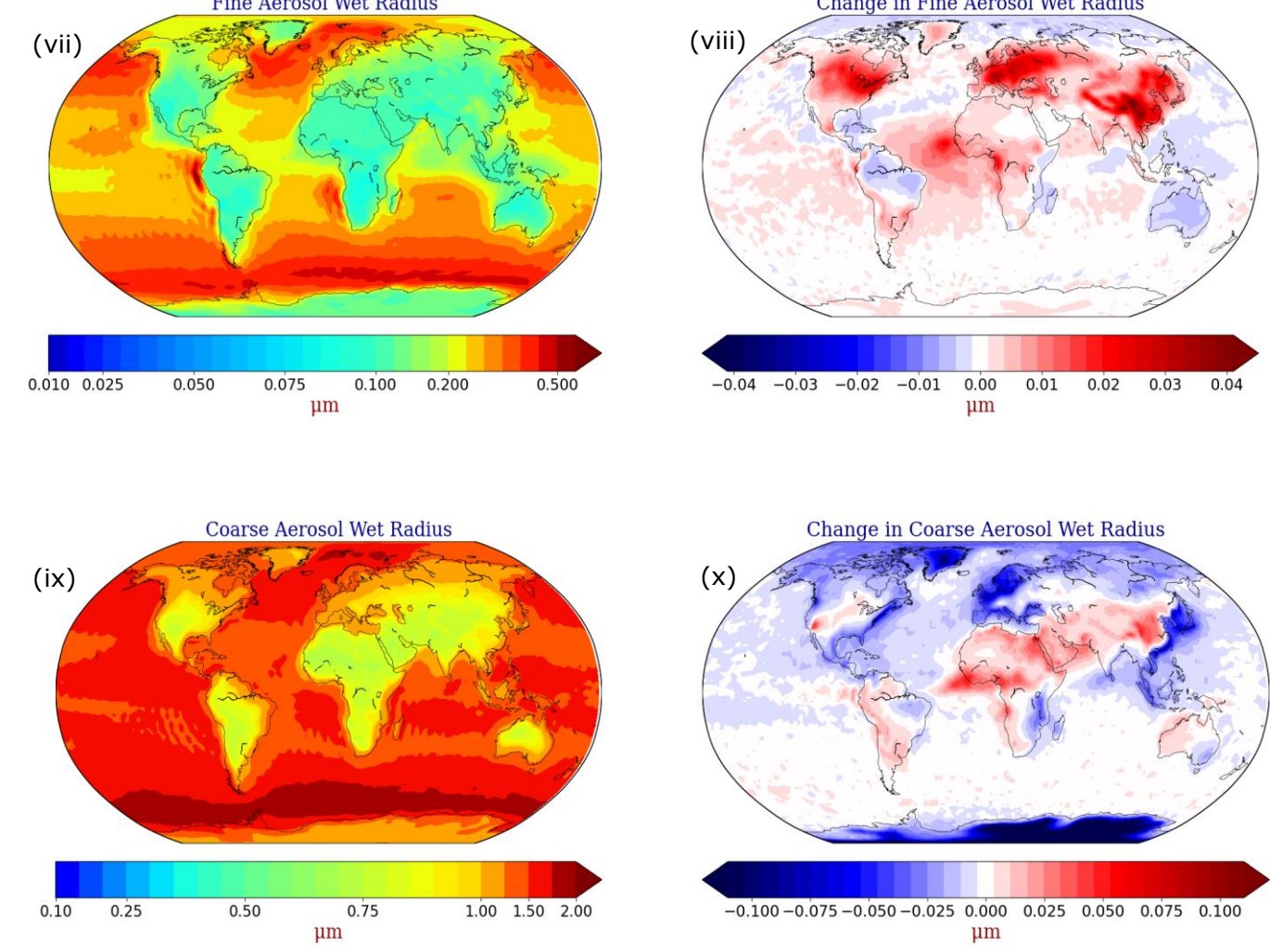

616

**Figure 4:** (i) Global mean maximum supersaturation, fine aerosol (iii) hygroscopicity and (v) wet radius, and coarse aerosol (vii) hygroscopicity and (ix) wet radius, as calculated by EMAC from the base case simulation at the altitude of 940 hPa. Absolute difference between base case and Nitrate Aerosol Free (NAF) sensitivity simulation in (ii) maximum supersaturation, fine aerosol (iv) hygroscopicity and (vi) wet radius, and coarse aerosol (viii) hygroscopicity and (x) wet radius at the altitude of 940 hPa. Red indicates higher values calculated by the base case simulation in the presence of $NO_3^-$ aerosols.

## 5.2    Number Concentrations of Aerosol and Activated Particles

Figure 5 shows the effect of $NO_3^-$ on the number concentration of fine and coarse aerosols between the base case and the 'NAF' sensitivity simulation, as well as the total aerosol population. The presence of $NO_3^-$ aerosols decreases the total aerosol number concentration over forests and polluted regions (see also Fig. 1d). This behavior is driven solely by the decrease in smaller particle sizes, as the effect is minimal for the coarser particles (Figs. 5ii & 5iv). The largest decrease is

calculated over East and South Asia (up to 1000 cm$^{-3}$ or 10%), while decreases of up to 200 cm$^{-3}$
on average (~10%) are found over Europe, the USA, and Central Africa. This effect is directly
related to the increased wet radius of the aerosol population (Fig. 4viii) over these regions and thus
to its depositional efficiency. In addition, coarse dust particles become more hygroscopic due to
interactions with $NO_3^-$ aerosols that increase in size, resulting in increased coagulation with the
smaller anthropogenic particles, which reduces their abundance.
The reduced aerosol number concentration in the presence of $NO_3^-$ can lead to a reduction of
particles that are also activated into cloud droplets. Such behavior can be seen in Figure 6, which
shows the effect of $NO_3^-$ on the number concentration of activated fine and coarse particles in
cloud droplets between the base case and the 'NAF' sensitivity simulation. The reduction in the
total number of activated cloud droplets is almost entirely due to the reduction in smaller size
particles (Figs. 6ii & 6iv). A reduction in the total number of activated droplets of up to 30 cm$^{-3}$ or
10% is observed over the USA, Amazon, Europe, Central Africa, and parts of the Middle East,
while this reduction reaches up to 100 cm$^{-3}$ (10%) over Southeast Asia, where the largest reductions
in aerosol numbers are also calculated (Fig. 4ii). In turn, these are the regions where the warming
effect of $NO_3^-$ aerosols on the calculated mean $RE_{aci}$ is strongest (Figure 3i). The small increase in
activated droplets (~ 10 cm$^{-3}$ or 1%) over Beijing, which concerns the fine mode particles, is most
likely because their number concentration decreases with increasing size. The high aerosol number
concentration there, which is the global maximum (Figure 5i), results in a hotspot of more readily
activated particles in the presence of $NO_3^-$. On the other hand, the CDNC decreases slightly over
the Sahara due to the more efficient deposition capacity of coarse dust particles due to their
interactions with nitrate aerosols, which is also reflected in the decrease in aerosol number (Fig.
6iv). Overall, the lower particle number in the presence of $NO_3^-$ aerosols hinders the ability of the
smaller anthropogenic particles to activate into cloud droplets, leading to a reduced cloud cover
and thus a reduced cloud albedo effect. Therefore, not only less LW radiation is absorbed, but
more importantly, less SW radiation is scattered back to space, resulting in an overall warming of
the net average $RE_{aci}$ for total $NO_3^-$ aerosols.

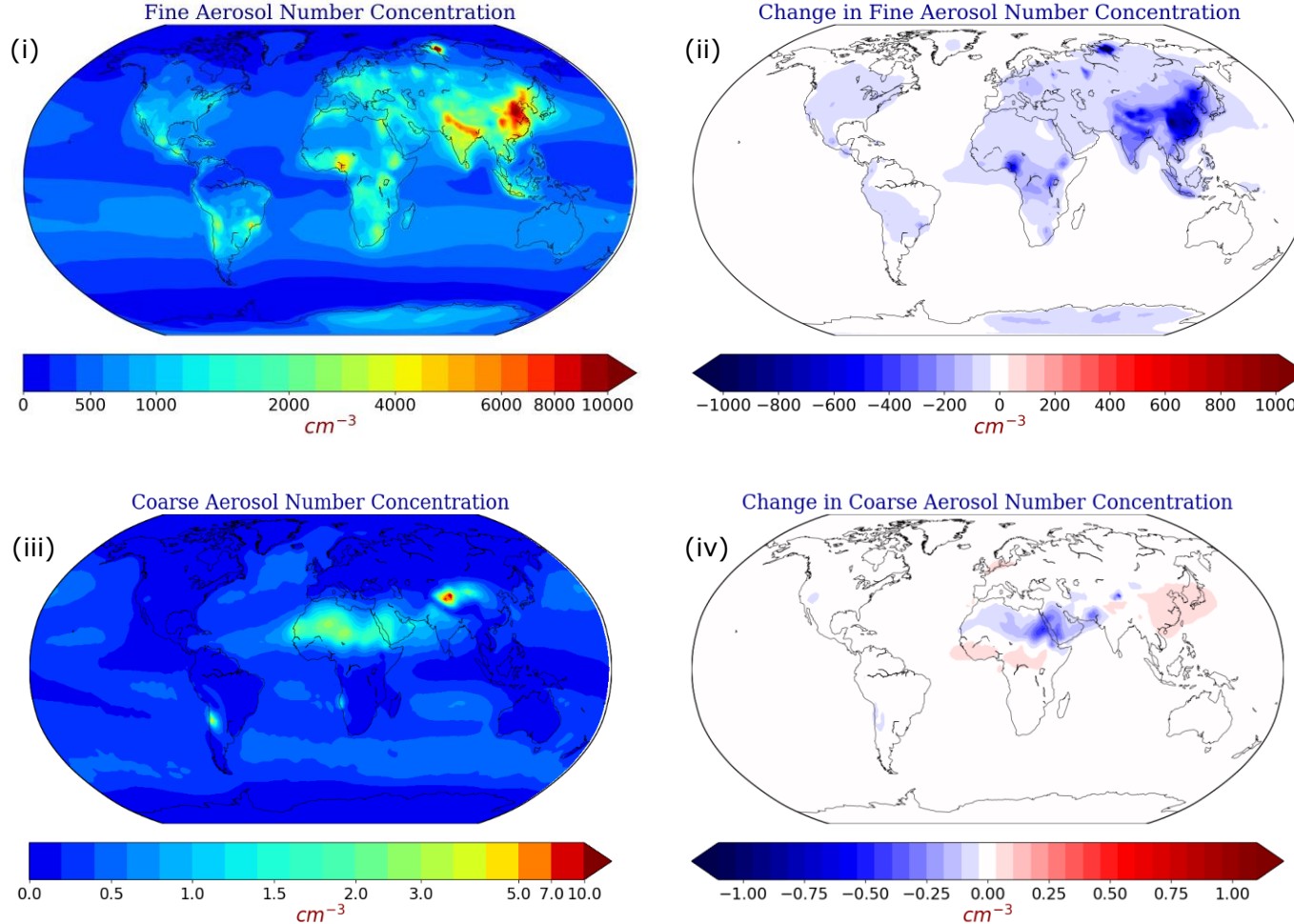


**Figure 5:** Global mean number concentration of (i) fine and (iii) coarse aerosols as calculated by EMAC from the base case simulation at the altitude of 940 hPa. Absolute difference between the base case and the Nitrate Aerosol Free (NAF) sensitivity simulation in the number concentration of (ii) fine and (iv) coarse aerosols at the altitude of 940 hPa. Blue indicates that number concentrations are lower in the presence of $NO_3^-$ aerosols.









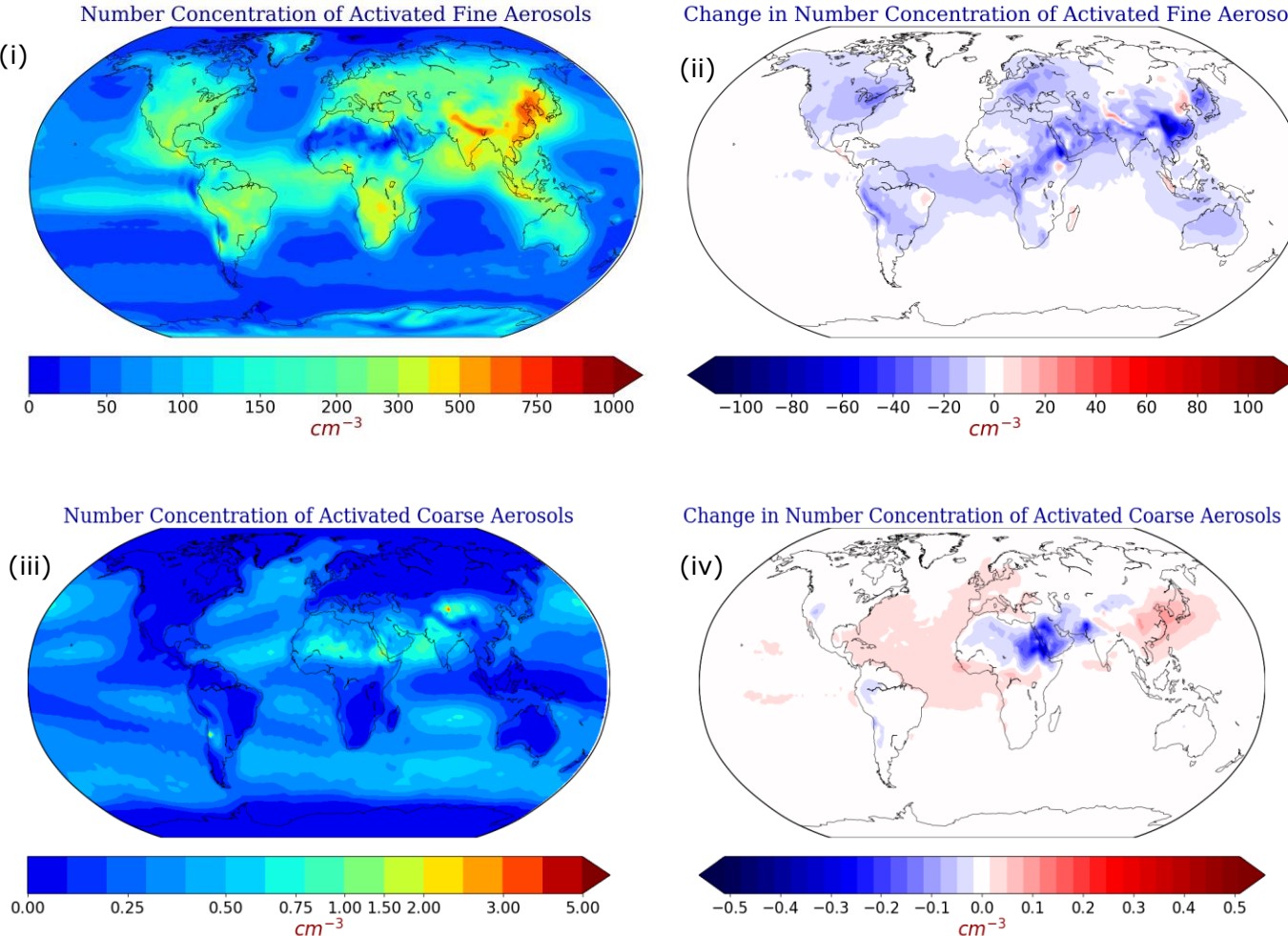

**Figure 6:** Global mean number concentration of activated (i) fine and (iii) coarse aerosols as calculated by EMAC from the base case simulation at the altitude of 940 hPa. Absolute difference between the base case and the Nitrate Aerosol Free (NAF) sensitivity simulation in the number concentration of activated (ii) fine and (iv) coarse aerosols at the altitude of 940 hPa. Blue indicates that number concentrations are lower in the presence of $NO_3^-$ aerosols.

# 6.    Conclusions and Discussion

This study presents the effects of interactions between mineral dust and $NO_3^-$ aerosols on the present-day global TOA radiative effect of the latter. We investigate how the presence of dust affects the radiative effect of $NO_3^-$ aerosols, both through aerosol interactions with radiation and separately with clouds ($RE_{ari}$ and $RE_{aci}$, respectively). Sensitivity simulations are also performed, varying both the mineral dust composition and its emissions, to assess their effect on the calculated $NO_3^-$ aerosol radiative effect.

It was found that the global average net $RE_{ari}$ of total $NO_3^-$ aerosols is -0.11 W/m$^2$, which is mainly due to the cooling from the shortwave part of the radiation spectrum due to scattering,

equal to -0.34 W/m$^2$. A warming from the longwave part of the spectrum due to absorption was found to be +0.23 W/m$^2$ on global average and was mainly located over regions with high concentrations of coarse NO$_3^-$ aerosols. SW cooling was also observed in these regions, but also over regions of high anthropogenic activity, mainly over the polluted northern hemisphere. The behavior of the RE$_{ari}$ was opposite when considering different sizes of NO$_3^-$ aerosols. Specifically, the coarse mode was responsible for 96% of the estimated warming in the LW part of the spectrum, but 15% of the estimated cooling in the SW part of the spectrum. On the other hand, the contribution of the fine mode to the LW warming was negligible, but it was the main contributor to the SW cooling, accounting for 85% of the net estimate. The sensitivity experiments revealed that the chemistry of the mineral dust is the most important factor in changing the estimated RE$_{ari}$ of the total NO$_3^-$ aerosols. In particular, LW warming is most affected by this assumption, being 52% weaker after assuming chemically inert dust emissions, while the SW cooling is reduced by 41% compared to the base case simulation, amounting to a net cooling of -0.09 W/m$^2$. A globally homogeneous ionic composition for mineral dust had a smaller effect in LW (22% decrease) and SW (21% decrease) but resulted in the same net estimate of -0.09 W/m$^2$. Halving the dust emissions resulted in weaker estimates for LW and SW by 17% and 21%, respectively, and the lowest overall net RE$_{ari}$ of -0.08 W/m$^2$. On the other hand, a 50% increase in dust emissions increased both LW warming and SW cooling by 17% and 9% respectively, resulting in a net cooling RE$_{ari}$ of -0.10 W/m$^2$, indicating the strong non-linear relationship of nitrate-dust interactions and how they affect the radiative effect estimates.

The global average net RE$_{aci}$ of total NO$_3^-$ aerosols was +0.17 W/m$^2$ due to the effect on the shortwave portion of the spectrum. This was found to be +0.27 W/m$^2$, while the cooling from the longwave part was -0.10 W/m$^2$. Spatially, the net RE$_{aci}$ is reversed compared to the net RE$_{ari}$ for total NO$_3^-$ aerosols, where regions responsible for a strong SW cooling of the RE$_{ari}$ contribute to a strong SW warming of the RE$_{aci}$ and vice versa. This is due to the fact that nitrate-dust interactions challenge the dominance of smaller particles over heavily polluted regions, reducing the reflectivity of warm cloud and thus having an opposite effect on the RE$_{aci}$. The sensitivity experiments again showed that the consideration of the mineral dust chemistry is the most important aspect for the calculation of the RE$_{aci}$ of the total NO$_3^-$ aerosols. When the dust was assumed to be chemically inert, the LW and SW estimates were up to 40% weaker, resulting in a net warming of +0.11 W/m$^2$. Assuming a homogeneous ion composition resulted in a smaller weakening of the estimates (up to 18%) and a net warming of +0.13 W/m$^2$. When dust emissions were halved, the LW cooling was reduced slightly more than in the base case, resulting in a net warming of +0.15 W/m$^2$. The 50% increase in dust emissions had the largest effect on LW behavior (10% increase), but surprisingly the net estimate (+0.14 W/m$^2$) was smaller than in the half-dust scenario. The reason for this is that the SW estimate did not increase but decreased by 8% due to the fact that in this scenario the increased nitrate burden causes increased competition for the available supersaturation and the effect of dust-nitrate interactions on the smaller aerosol populations is not as emphasized as in the base case.

The total NO$_3^-$ aerosol RE$_{aci}$ shows a positive sign, which is attributed to a reduced cloud albedo effect. More specifically, although the presence or absence of NO$_3^-$ aerosol in the atmosphere did not significantly affect the total available maximum supersaturation, it did alter both the hygroscopicity and wet radii of the aerosols. In the presence of NO$_3^-$, the hygroscopicity of aerosols

over deserts was increased by up to an order of magnitude, leading to an increase in their wet radius of up to 10%, with an even larger increase of up to 40% for smaller particles over urban regions. Therefore, in the presence of $NO_3^-$ aerosols, the depletion of fine particles by coagulation with coarser particles (i.e., mineral dust) is enhanced and further increases the size of the coarse particles. The reduction in the number of aerosols is up to 10% in some regions, with maximum reductions calculated over Southeast Asia. This reduction in the number of fine aerosols leads to a reduction in the number of cloud droplets activated by fine aerosols (also up to 10%), which would otherwise have absorbed more outgoing longwave radiation and, more importantly, scattered more incoming shortwave radiation. Thus, the reduced cloud albedo effect leads to a cooling in the longwave part of the spectrum, which is offset by a strong warming in the shortwave part, overall resulting in a net warming of the atmosphere.

The chemistry-climate model simulations presented here suggest that $NO_3^-$ aerosol-radiation interactions lead to a net effect of -0.11 $W/m^2$ (cooling) driven by fine $NO_3^-$ aerosol, while $NO_3^-$ aerosol-cloud interactions lead to a net effect of +0.17 $W/m^2$ (warming) driven mainly by coarse mode $NO_3^-$ aerosol.

**Code and Data Availability**

The usage of MESSy (Modular Earth Submodel System) and access to the source code is licensed to all affiliates of institutions which are members of the MESSy Consortium. Institutions can become a member of the MESSy Consortium by signing the "MESSy Memorandum of Understanding". More information can be found on the MESSy Consortium website: http://www.messy-interface.org (last access: 22 May 2024). The code used in this study has been based on MESSy version 2.55 and is archived with a restricted access DOI (https://doi.org/10.5281/zenodo.8379120, The MESSy Consortium, 2023). The data produced in the study is available from the authors upon request.

**Acknowledgements**

This work was supported by the project FORCeS funded from the European Union's Horizon 2020 research and innovation program under grant agreement No 821205. JFK was funded by the National Science Foundation (NSF) Directorate for Geosciences grants 1856389 and 2151093. The work described in this paper has received funding from the Initiative and Networking Fund of the Helmholtz Association through the project "Advanced Earth System Modelling Capacity (ESM)". The authors gratefully acknowledge the Earth System Modelling Project (ESM) for funding this work by providing computing time on the ESM partition of the supercomputer JUWELS (Alvarez, 2021) at the Jülich Supercomputing Centre (JSC).

## Competing Interests

At least one of the (co-)authors is a member of the editorial board of Atmospheric Chemistry and Physics.

## Author Contributions

AM and VAK wrote the paper with contributions from KK, APT, JFK, MK, and AN. VAK planned the research with contributions from APT, MK and AN. AM, KK and VAK designed the methodology for the radiative effect calculations. AM performed the simulations and analyzed the results, assisted by VAK and APT. All the authors discussed the results and contributed to the paper.

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
