# Peer review of "Impact of mineral dust on the global nitrate aerosol direct and indirect radiative effect"

_EGUsphere, 2024_

## Author Comment (AC1)

**Authors' response to comments made by anonymous reviewer #1:**

**Summary**

*This manuscript presents a modeling effort to quantify the impact of mineral dust on radiative forcing effect of nitrate aerosol. Nitrate was believed to be a cooling aerosol although great uncertainties remain within the estimated radiative forcing. As the authors mentioned, nitrate is probably going to play a more important role due to the decrease in sulfate. In addition, mineral dust was known to affect the formation of nitrate aerosol through thermodynamic equilibrium and heterogeneous reactions. I strongly agree that a thorough investigation of the interaction between dust and nitrate and the associated radiative forcing effect will help to improve the understanding of climate change. Therefore, this study focused on an important and interesting topic and applied a proper modeling method. In general, the manuscript is well written with a clear description of the objective and well-organized discussions of the results. However, there are a few major issues regarding the modeling method which requires more details to help better justify and support the results and conclusion of this study. I would recommend a major revision to address these issues before a final decision could be made regarding the acceptance of the submission. Please find the major and minor comments below.*

We would like to thank the reviewer for his/her thoughtful review and positive response. Below is a point-by-point response (in black) to the major and minor comments (in blue).

**Major Comments**

1. *A main issue is that it seems this study didn't consider dust heterogeneous chemistry which may promote conversion from NOx to nitrate on the surface of dust particles. The "physicochemical interactions of mineral dust particles with gas and aerosol tracers" in this study refers to the thermodynamic equilibrium between gas-phase HNO3 and particle phase nitrate if I understand the manuscript correctly. I am not quite sure which one of these two process, heterogeneous chemistry or thermodynamic equilibrium, plays a major role in dominating the production of nitrate in the present of dust as there is no such demonstration in this work. Similarly, dust heterogeneous chemistry also promotes conversion from gas phase SO2 to sulfate which may further affect the thermodynamic equilibrium of HNO3. Therefore, as the topic focused on "impact of dust on the global nitrate", I would recommend the authors to include a brief discussion to explain that omitting dust heterogeneous chemistry may or may not affect the conclusion of this study.*

In addition to nitrate production on dust particles by thermodynamic equilibrium between gas-phase $HNO_3$ and particulate nitrate, this study also considers production via heterogeneous chemistry by hydrolysis of N2O5. It has been shown that this chemical formation pathway is the most dominant for heterogeneous nitrate production (Seisel et al., 2005; Tang et al., 2012), while others such as $NO_2$ oxidation do not show such high yields, although they are also important during dust pollution events over polluted regions (Li et al., 2024). The same is true for heterogeneous $SO_2$ oxidation and therefore it would not significantly affect the thermodynamic equilibrium of $HNO_3$ under most conditions. Consideration of sulphate production by heterogeneous chemistry

could theoretically result in reduced amounts of particulate nitrate in some cases due to acidification of dust particles, which inhibits partitioning of $HNO_3$ to the aerosol phase (Nenes et al., 2020). More information on the heterogeneous nitrate production considered in this study has been added to the relevant part of Section 1 in the revised version. In addition, a disclaimer regarding the omission of full consideration of heterogeneous chemistry and its potential impact on our results has been added to Section 2.2 in the revised version.

2. *The second main issue with the study is a lack of discussion about uncertainties of the REari and REaci estimations. Radiative forcing effect is a complex index calculated based on a series of model simulated variables, and it may subsequently inherit the associated uncertainties. For example, how were the model performances for simulating dust emission and size distribution, nitrate concentration, aerosol vertical distributions? These variables will significantly affect the estimations of REari and REaci, and a clear demonstration of modeling biases for these variables will help audiences to better understand the radiative effects quantified by the modeling system. Especially, many climate models represent formation of nitrate in a very simplified manner. Therefore, I am very interested to see how well the model used in this study can simulate mass concentration of nitrates as evaluated against observations.*

We acknowledge the lack of information on the ability of the model to accurately simulate the amounts of nitrate and dust aerosol in the atmosphere and agree with the reviewer that such an addition would indeed help readers to better assess the credibility of the study's conclusions. EMAC is routinely evaluated against ground-based, aircraft, and satellite observations of aerosol concentrations and composition, aerosol optical depth, acid deposition, gas-phase mixing ratios, cloud properties, and meteorological parameters (Tsimpidi et al., 2016, 2017; Karydis et al., 2016; Karydis et al., 2017; Bacer et al., 2018; Pozzer et al., 2022). Here, we have included in the supplementary material the zonal profiles for coarse and fine nitrate aerosols and for the mineral ions present in dust particles. We have also included a comparison of model results for surface concentrations of $PM_{2.5}$ nitrate aerosols with observations from measurement networks in the most active regions of the polluted northern hemisphere (EANET, EMEP, EPA & IMPROVE).

**Minor Comments**

1. *Table 1 talks about details of simulation configuration, so it might be better to move it to section 2.*

Yes, this is true, as it essentially summarizes the last paragraph of Section 2.1. It has therefore been moved to the end of that section in the revised manuscript.

2. *It's necessary to briefly describe the difference in ionic composition between Karydis et al. (2016) and global homogeneous setting.*

A detailed description of the ionic composition of both cases, namely Karydis et al. (2016) and Sposito (1989), is mentioned at the end of Section 2.1, along with the details of the other sensitivity

simulations performed. In the revised version, we have also added footnotes to Table 1 to make this information more visible to the reader.

3.  *Better use math symbol instead of letter for "x".*

This has been corrected in the revised version, where the grid resolution is entered as a mathematical equation instead of plain text.

4.  *These "mineral ions" are treated as individual particles or as supplements of dust particle?*

These mineral ions are treated as individual species that are part of the aerosol in each size mode and are assumed to be well mixed with the rest of the aerosol species considered (i.e., dust, black carbon, organics, inorganic ions). In total, EMAC considers 7 particles described by lognormal size modes (four hydrophilic and three hydrophobic). The aerosol composition within each mode is uniform in size (internally mixed) but can vary between modes (externally mixed). This information has been added to the revised manuscript.

5.  *Does "chemically inert" mean the gas-phase HNO3 adsorbed onto the surface of dust particle will not partition into nitrate through equilibrium with NVCs?*

In the base case simulation, nitrate is formed by the rapid uptake of $HNO_3$ by dust particles due to simple acid–base interactions with the NVCs. In the "chemically inert" case, these interactions do not occur because there are no NVCs in the dust composition and therefore $HNO_3$ remains in the gas phase. This information is now included in the revised version.

6.  *Do you mean 94% of emitted dust mass is treated as "bulk dust" in the model? The particle size distributions of dust and other ions are the same?*

Yes, exactly. In this sensitivity, 94 % of the emitted dust mass is treated as a bulk species in the model. The size distribution of the emitted dust mass remains the same as before, only the ionic composition is different. This clarification has been added in the revised version.

7.  *Please include the function here to help illustrate the partition process and how it is represented in the model.*

The equation describing the diffusion gas flux on a single particle surface, which is essentially the amount of gas that can kinetically condense on it within a time step, as described in Vignati et al. (2004), has been added to the text at the beginning of Section 2.2 in the revised version.

8.  *Freshly emitted species are usually NOx instead of HNO3.*

This is of course the case. It was not our intention to suggest that $HNO_3$ is a primary pollutant rather than a transported one, as stated in the figure caption. A reference to 'freshly emitted nitric acid' has been corrected to 'freshly formed nitric acid'. We understand that the small dark/green arrows could possibly give the impression of a direct emission to the atmosphere, but our aim was

to imply only transport (they are also shown above the surface), as fresh emissions are represented by the thicker arrows coming from the 3 different emission categories at the surface.

*9.  Please specify how exactly nitrate formation was turned off. Was gas phase HNO3 still condense but no nitrate was produced, or HNO3 is the end product of nitrous oxide?*

For these sensitivity simulations we have assumed that no $HNO_3$ is produced from $NO_2$ oxidation and $N_2O_5$ hydrolysis. Therefore, in the model simulations with nitrate formation completely switched off, there is no $HNO_3$ in the atmosphere to condense in the aerosol via equilibrium partitioning and form nitrate. A clarification has been added at the beginning of Section 2.3.1 in the revised version.

*10. If HNO3 was forced to condense only on fine mode, will this lead to a lower level of sulfate formation since the model need to keep the equilibrium? Will there be any non-linear response of other aerosols such as sulfate by tuning off condensation? If we set up a 4$^{th}$ simulation by forcing HNO3 to condense only on coarse mode, can we estimate the radiative effect as: FfineNO3,ari = F1,ari – F4,ari ?*

The amount of nitric acid that would otherwise condense on the coarse mode remains in the gas phase. Since sulfuric acid has an extremely low vapor pressure, it will partition completely into the aerosol phase anyway. So, such a change in $HNO_3$ partitioning cannot lead to a non-linear response of sulfate aerosols. Finally, yes, theoretically, the direct radiative effect of fine nitrates can be calculated in a setup like the one proposed here, since it is a similar methodology to the one used in this study to calculate the coarse mode radiative effect. However, by allowing $HNO_3$ to condense only on the coarse mode, the results are more prone to errors due to kinetic limitations on the coarse particles, since the $HNO_3$ left in the gas phase by the missing condensation in the fine mode will be available to condense on the coarse particles, leading to an overestimation of nitrate.

*11. As there was no aerosol-cloud interaction, does it mean the simplest cloud scheme apply a prescribed aerosol configuration? Can it properly reproduce cloud over the study period?*

The cloud scheme used for all instances of the direct radiative effect calculations is a statistical cloud cover scheme using prognostic equations for the different water phases and distribution moments as described in Tompkins (2002) and also Roeckner et al. (2006). The bulk microphysics of the scheme follows the methodology of Lohmann and Roeckner (1996), where the cloud droplet number concentration (CDNC) is empirically related to the sulfate aerosol mass and more specifically its monthly mean values derived from the sulfur cycle in ECHAM. The detailed equations for the marine and continental CDNC can be found in Lohmann and Roeckner (1996), and more detailed information on the aerosol sulfate mass can be found in Boucher and Lohmann (1995) as well as Feichter et al., (1996). This information has now been added to the revised version. This particular scheme has indeed been proved to be able to accurately simulate cloud cover, although this particular aspect is not of major importance for the calculation of aerosol-radiation interactions.

*12. Model configuration for cloud scheme is a little confusing here, line#313 mentions FN,ari is calculated using method in sec2.3.1 which applied the simplest cloud scheme as mentioned in line#294, but it seems in order to estimate aci, another set of cloud scheme was used.*

As explained in Section 2.3.2, in order to calculate the indirect effect, we first estimate the feedback radiative effect of nitrates using two additional simulations for each sensitivity case, using the more advanced cloud scheme (described in Section 2.3.2). Indeed, the feedback and direct radiative effects were calculated with different cloud schemes, because for the latter it was necessary to eliminate any climatological influences, but for the former it was essential to include them. However, since the feedback effect could be considered as an estimate that includes both the direct and indirect effects, in order to isolate the indirect effect, we had to subtract the direct effect, as correctly calculated with the three initial simulations (Section 2.3.1).

*13. A table showing REari reported in these references would be helpful to better demonstrate the comparison.*

This is an excellent suggestion. A table comparing the estimates of the REari for total nitrate aerosols between this study and those referred to at the beginning of Section 3.1 has been added to the supplementary material of the manuscript, and the reader is referred to it in the revised version.

*14. Please explain why there is a strong warming dot over Sahara in Fig.2(v).*

The presence of this localized warming over the Sahara in the SW part of the spectrum is related to the interactions of nitrate aerosols with dust particles, in combination with the region itself. Because the underlying desert surface is so bright, its absorption is less than that of the particles above it at these wavelengths, which means that the surface of the desert can scatter radiation more effectively than the particles. This is amplified by particle growth there, as seen by the increase in the coarse mode wet radius over the Sahara in the presence of nitrates, which also means an increase in the absorption cross section of the particles. This leads to positive forcing in parts of the region and weak negative forcing in other parts. The explanation in the text has been changed in the revised version to better describe this interplay to the reader.

*15. It's a very interesting point that nitrate REari seems insensitive to dust load but Table 2 suggested that interaction between nitrate and dust has a significant impact on coarse mode aerosols' LW and SW forcing. It's better to include more detailed discussions in this paragraph to explain why "nitrate-dust interactions are not linearly correlated", and why "a given increase or decrease in dust emissions does not lead to an analogous change in nitrate aerosol level".*

In fact, the behavior of the nitrate REari is not insensitive to dust loading, as changes in this also led to altered estimates of the LW & SW forcing (Table 2). However, the changes are indeed more pronounced when considering the inclusion of dust chemistry. It is a good idea to include some more discussion of why this non-linear behavior exists, so that the reader has a broader understanding of the interactions between dust and nitrates. Therefore, an additional paragraph has been added at the end of Section 3.2 in the revised manuscript to cover this aspect. In short, the

amount of HNO$_3$ present over dust aerosol surface is mainly the limiting factor for nitrate production (due to adsorption on dust particles) than the amount of dust itself. Furthermore, in cases where more dust is present in the atmosphere, its increased removal rates by wet deposition and/or coagulation led to non-analogous increases in nitrate production.

*16. What is the "advanced cloud scheme" ?*

The advanced cloud scheme refers to the one used in the two additional simulations from which the nitrate feedback radiative effect was estimated and is described at the end of Section 2.3.2. This clarification has been added in the revised version. More specifically, it is the scheme of Lohmann and Ferrachat (2010), which uses prognostic equations for the water phases and bulk cloud microphysics. It also uses the empirical cloud cover scheme of Sundqvist et al. (1989). In addition, it uses the CDNC activation scheme of Morales and Nenes (2014) for aerosol activation, which includes the adsorption activation of mineral dust as described in Karydis et al. (2017). Finally, the scheme of Barahona and Nenes (2009) is used for the ICNC activation, which calculates the ice crystal size distribution through heterogeneous and homogeneous freezing and ice crystal growth as described in Bacer et al. (2018).

**References**

[revised manuscript text omitted]

---

## Author Comment (AC2)

**Authors' response to comments made by anonymous reviewer #2:**

**Summary**

*The manuscript titled "Impact of mineral dust on the global nitrate aerosol direct and indirect radiative effect" by Milousis et al. investigated the radiative effects of nitrate on dust by using a climate model, including the aerosol-radiation interactions and aerosol-cloud interactions. Nitrate chemistry on dust is implemented in the EMAC model, simulations were conducted based on the base case and several sensitivity simulations. In general, the logic of the study is explicit and the organization of the manuscript structure is clear. However, the study lacks of necessary evaluation of the simulation results and thus the results could be subject to high uncertainties.*

We would like to thank the reviewer for his/her thoughtful review and positive response. Below is a point-by-point response (in black) to his/her comments (in blue).

**Major Comments**

1. *There is no comparison between model results and observations, e.g. mass concentrations of nitrate, dust (PM10), aerosol number concentrations. There are plenty of observational datasets or literature values available. The lacking of constraints from observational data will reduce the credibility of model simulations.*

We agree with the reviewer that including comparisons with observational data to demonstrate the ability of the model to provide realistic estimates of both aerosol concentrations and cloud droplet numbers will help to increase the credibility of the study's findings. For this reason, we have now included in the supplementary material a comparison of the model results for surface mass concentrations of $PM_{10}$ aerosols with observations from measurement networks in the polluted Northern Hemisphere (EANET, EMEP & IMPROVE). In addition, we have also included a comparison between the CDNCs simulated by the model and those measured in a variety of regions across the world (continental, polluted and clean marine) over different time periods and altitudes, as found in Karydis et al., (2017) and all relevant references therein. The reader is made aware of this content at the end of Section 2.1 in the revised version.

2. *Figure 2 & Line 371 – 377: In Figure 2v and 2vi, the TOA SW REari of coarse nitrate is much stronger than that of fine nitrate, it seems unreasonable as fine nitrate dominates the total nitrate, especially in East Asia.*

This is an excellent point, which helped us discover an error in the simulation where $HNO_3$ should only condense on the fine mode (i.e., the simulations where the coarse mode was excluded). The error came from the thermodynamic calculations, where only negligible amounts of $HNO_3$ actually condensed on the fine mode, resulting in unrealistically low fine nitrate concentrations and thus such weak estimates of the radiative effect. This error affected the contribution of the fine and coarse modes to the direct radiative estimate of total nitrate (namely $F_{fN,ari}$ & $F_{cN,ari}$) and not the estimate of the total nitrate aerosol itself ($F_{N,ari}$) in Section 2.3.1. This is because only the

quantity $F_{3,ari}$ (calculated taking into account all aerosol components except coarse $NO_3^-$) was incorrectly calculated. Therefore, this error was not transferred to the calculation of the indirect radiative estimate, since only the quantity $F_{N,ari}$ was used for this (Section 2.3.2). We have now corrected this error by ensuring that for simulations where the coarse mode is removed from the aerosol load, the thermodynamic calculations for $HNO_3$ condensation are performed correctly and the condensed $HNO_3$ is only transferred to the fine mode. For this reason, we have performed 5 new simulations (1 for each sensitivity case) where these conditions apply. As a result, Figure 2 and Table 2 have been updated with the correct results for the fine and coarse mode estimates for the direct radiative effect. In addition, Sections 3.1 and 3.2 have been thoroughly revised to incorporate the new results.

3. *Figure 4: the kappa values of fine aerosol over the continents are mostly lower than 0.04 (iii), which are incorrect. Even considering the mixing between dust and anthropogenic emissions, the hygroscopicity of aerosols couldn't be so weak.*

   While the kappa values of the fine aerosol population appear to be low, particularly over the dust belt zone, this is largely due to the higher proportion of insoluble fine aerosols present there. This is also observed over other regions with similarly low fine aerosol hygroscopicity (South Africa, South America and Western U.S). Furthermore, the estimates of aerosol kappa values at 940 hPa are broadly in agreement with the findings of Pringle et al., (2010). We have included the model estimates for the global insoluble fractions of the fine and coarse aerosol populations in the revised supplementary material. The reader is referred to this in the relevant part of Section 5.1 in the revised manuscript.

4. *Line 195 – 197: Na+, K+, Ca2+ and Mg2+ constituted 100% of bulk dust? How are the anions treated?*

   No, this is not the case. The composition of the emitted mineral dust consists of a bulk component, which accounts for  94% of the emitted flux, and the remaining 6% represents the mass fractions of the mineral cations. No anions are considered to be explicitly emitted as part of the emitted dust flux.

5. *Line 222 – 223: How is the delinquencies of salts treated in the model under different relative humidity?*

   In our model, the deliquescence of salts under different relative humidities is treated according to the Mutual Deliquescence Relative Humidity (MDRH) approach of Wexler and Seinfeld (1991), as described in Fountoukis and Nenes (2007). More specifically, each individual salt has a certain threshold, the DRH, above which its phase transition from solid to liquid occurs. However, in the presence of a multicomponent mixture, it is the MDRH that determines the humidity value above which all salts in the mixture are considered to be saturated. The MDRH is below the DRH of all the pure solids in the mixture. As the RH over a wet particle decreases, the aerosol may not crystallize below the MDRH but instead remain in a state where it consists of an aqueous solution that is supersaturated with dissolved salts. This state is called metastable and is the state considered in our study by the ISORROPIA-lite thermodynamic model (Kakavas et al., 2022; Milousis et al.,

2024). This information has been added in the revised version of the manuscript in Section 2.2 right after the deliquescence chemical reactions.

6.  *Section 4: Why the radiative effects from Aerosol-Cloud Interactions are not separated for fine and coarse nitrate?*

This is a valid question. As explained at the beginning of Section 2.3.2 on the REaci calculation methodology, we estimate it in this way because it is essential to include feedbacks from different climatological conditions. In particular, since climatology plays a crucial role in aerosol-cloud interactions, the simulation of a "fine-only $NO_3^-$ atmosphere", as done for the REari calculations, would produce a climatological scenario that would lead to inaccurate estimates of the feedback radiative effect of nitrate aerosol. This is because coarse-mode $NO_3^-$ is strongly associated with cations in mineral dust particles (Karydis et al., 2016), making them quite effective as CCN (Karydis et al., 2017). Separating of the direct radiative effect between fine and coarse nitrate is a simpler task that provides realistic results because of two conditions. First, not only are the aerosol-cloud interactions switched off (by not considering any specific parameterization for aerosol activation besides the cloud cover prognostic equations, as described at the end of Section 2.3.1), but also any potential uncertainties due to different climatological conditions are eliminated in this case, as explained in Section 2.3.1.

**References**

Fountoukis, C. and Nenes, A.: ISORROPIA II: a computationally efficient thermodynamic equilibrium model for $K^+$–$Ca^{2+}$–$Mg^{2+}$–$NH_4^+$–$Na^+$–$SO_4^{2-}$–$NO_3^-$–$Cl^-$–$H_2O$ aerosols, Atmos. Chem. Phys., 7, 4639–4659, https://doi.org/10.5194/acp-7-4639-2007 , 2007.

Kakavas, S., Pandis, S. N., and Nenes, A.: ISORROPIA-Lite: A Comprehensive Atmospheric Aerosol Thermodynamics Module for Earth System Models, Tellus Series B-Chemical and Physical Meteorology, 74(1), 1-23, https://doi.org/10.16993/tellusb.33 , 2022.

Karydis, V. A., Tsimpidi, A. P., Pozzer, A., Astitha, M., and Lelieveld, J.: Effects of mineral dust on global atmospheric nitrate concentrations, Atmospheric Chemistry and Physics, 16(3), 1491-1509, https://doi.org/10.5194/acp-16-1491-2016 , 2016.

Karydis, V. A., Tsimpidi, A. P., Bacer, S., Pozzer, A., Nenes, A., and Lelieveld, J.: Global impact of mineral dust on cloud droplet number concentration, Atmospheric Chemistry and Physics, 17(9), 5601-5621, https://doi.org/10.5194/acp-17-5601-2017 , 2017.

Milousis, A., Tsimpidi, A. P., Tost, H., Pandis, S. N., Nenes, A., Kiendler-Scharr, A., and Karydis, V. A.: Implementation of the ISORROPIA-lite aerosol thermodynamics model into the EMAC chemistry climate model (based on MESSy v2.55): implications for aerosol composition and acidity, Geoscientific Model Development, 17(3), 1111-1131, https://doi.org/10.5194/gmd-17-1111-2024 , 2024.

Pringle, K. J., Tost, H., Pozzer, A., Pöschl, U., and Lelieveld, J.: Global distribution of the effective aerosol hygroscopicity parameter for CCN activation, Atmos. Chem. Phys., 10, 5241–5255, https://doi.org/10.5194/acp-10-5241-2010 , 2010.

Wexler, A. S., & Seinfeld, J. H.: Second-generation inorganic aerosol model. *Atmospheric Environment. Part A. General Topics*, *25*(12), 2731-2748, https://doi.org/10.1016/0960-1686(91)90203-J, 1991.